# How Does Sequence Modeling Architecture Influence Base Capabilities of Pre-trained Language Models? Exploring Key Architecture Design Principles to Avoid Base Capabilities Degradation

**Xin Lu**[1], **Yanyan Zhao**[1,*], **Si Wei**[2,*], **Shijin Wang**[2], **Bing Qin**[1], **Ting Liu**[1]

[1]Research Center for Social Computing and Interactive Robotics, Harbin Institute of Technology
[2]iFLYTEK Co., Ltd
[1]{xlu, yyzhao, qinb, tliu}@ir.hit.edu.cn
[2]{siwei, sjwang3}@iflytek.com

## Abstract

Pre-trained language models represented by the Transformer have been proven to possess strong base capabilities, and the representative self-attention mechanism in the Transformer has become a classic in sequence modeling architectures. Different from the work of proposing sequence modeling architecture to improve the efficiency of attention mechanism, this work focuses on the impact of sequence modeling architectures on base capabilities. Specifically, our concern is: How exactly do sequence modeling architectures affect the base capabilities of pre-trained language models? In this work, we first point out that the mixed domain pre-training setting commonly adopted in existing architecture design works fails to adequately reveal the differences in base capabilities among various architectures. To address this, we propose a limited domain pre-training setting with out-of-distribution testing, which successfully uncovers significant differences in base capabilities among architectures at an early stage. Next, we analyze the base capabilities of stateful sequence modeling architectures, and find that they exhibit significant degradation in base capabilities compared to the Transformer. Then, through a series of architecture component analysis, we summarize a key architecture design principle: A sequence modeling architecture need possess full-sequence arbitrary selection capability to avoid degradation in base capabilities. Finally, we empirically validate this principle using an extremely simple Top-1 element selection architecture and further generalize it to a more practical Top-1 chunk selection architecture. Experimental results demonstrate our proposed sequence modeling architecture design principle and suggest that our work can serve as a valuable reference for future architecture improvements and novel designs.

## 1 Introduction

Recent research has discovered that pre-trained language models [35, 12, 6, 30], represented by Transformer [47], possess strong base capabilities and can achieve excellent performance in **language modeling**, **few-shot learning**, etc. Delving into the specific architecture design of Transformer, its self-attention mechanism is widely regarded as one of the key components behind its success and has since become a classic in sequence modeling architectures.

---

* Email corresponding.

39th Conference on Neural Information Processing Systems (NeurIPS 2025).

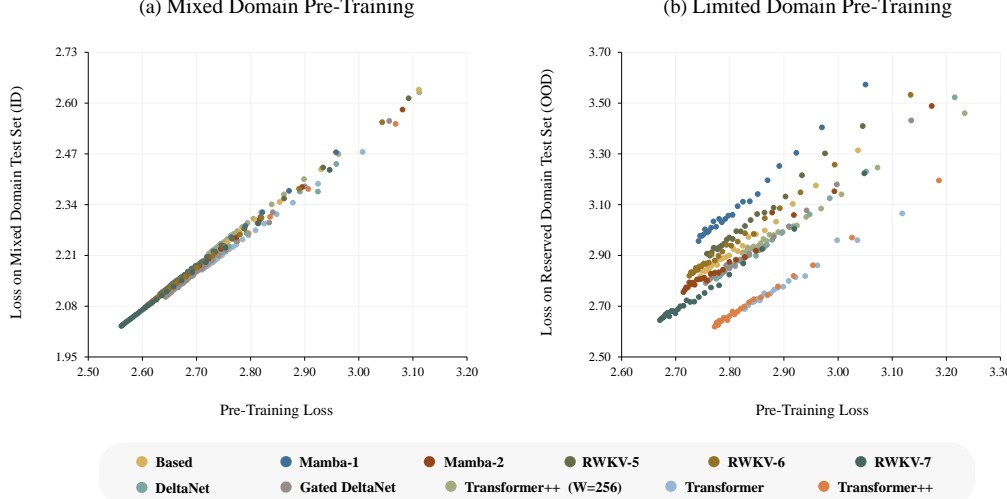

Figure 1: Language modeling test results of various sequence modeling architectures under two pre-training settings. (Model parameters≈110M, pre-trained tokens=100B and sequence length=2k)

However, due to its quadratic time complexity, the self-attention mechanism has long been plagued by high computational costs in long-sequence scenarios. To improve the efficiency of sequence modeling architectures, numerous novel stateful sequence modeling architectures have recently emerged, such as Mamba [16, 11], RWKV [31, 32, 33], Gated DeltaNet [51]. These architectures primarily inherit stateful modeling mechanisms of linear attention [23] and linear RNNs [28], offering advantages in time and space efficiency over self-attention. Additionally, they introduce new mechanisms like data-dependent decay and delta rule to enhance the expressive power of linearized modeling.

Although experimental results demonstrate that these stateful sequence modeling architectures can match or even surpass Transformer in performance while maintaining significant efficiency advantages, some studies have noted their deficiencies in specialized capabilities such as retrieval [49], copy [21], associative recall [1], and dynamic programming [50], supported by empirical validation on synthetic datasets and tasks. While these studies focus on specific issues, they are highly insightful and directly raise our new questions about sequence modeling architectures:

*Are the limitations of stateful sequence modeling architectures not only confined to specialized capabilities but also present in base capabilities?*

*What architecture factors truly influence base capabilities?*

*What design principles can prevent the degradation of base capabilities?*

To address these questions, we first point out that existing research on sequence modeling architectures typically adopts the same mixed domain pre-training settings used in large model development. While beneficial for practical applications, these settings are detrimental to architecture analysis, as they turn base capability tests (e.g., language modeling) into in-distribution evaluations, failing to reveal differences in base capabilities during early pre-training stages. To address this, we propose a limited domain pre-training approach, **employing out-of-distribution language modeling performance to measure base capabilities**, successfully uncovering significant differences in base capabilities among architectures at an early stage.

Next, under this setting, we analyze the base capabilities of stateful sequence modeling architectures like Mamba, RWKV, Gated DeltaNet. Our findings reveal that these architectures exhibit notable degradation in base capabilities compared to Transformer, confirming that stateful sequence modeling architectures suffer from deficiencies not only in specialized capabilities but also in base capabilities.

We then investigate the architecture factors that truly impact base capabilities. Through ablation studies on the Mamba family of architectures and analyses of common sequence modeling factors, we identify that mechanisms like data-dependent decay, convolution and position encoding only affect convergence speed rather than base capabilities. Conversely, we determine that full-sequence visibility, real relation calculation and non-uniform distribution are critical architecture factors

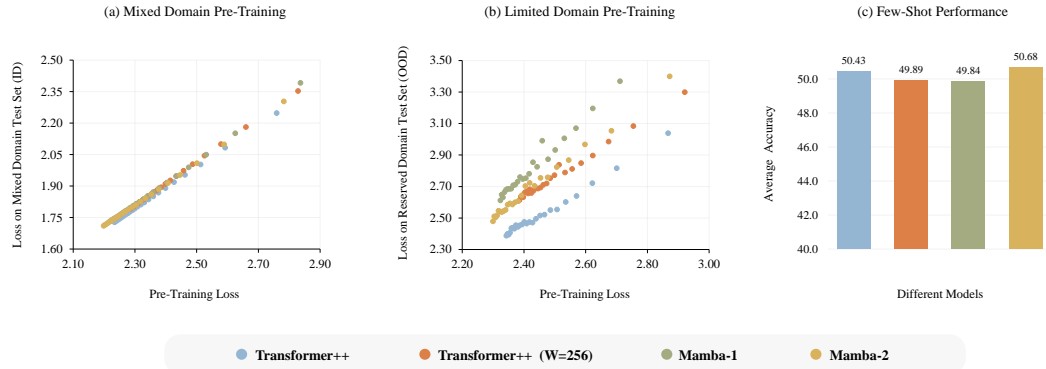

Figure 2: The Illustration include: (a) and (b) Language modeling test results of various sequence modeling architectures under two pre-training settings. (c) Few-shot learning performance results of these architectures. (Model parameters≈1.3B, pre-trained tokens=100B and sequence length=2k)

influencing base capabilities. Based on this, we summarize the principle that **"a sequence modeling architecture need to possess full-sequence arbitrary selection capability"** as the key design rule to avoid degradation in base capabilities.

Finally, we validate this principle using an extremely simple Top-1 Element Selection architecture, experimentally confirming its ability to achieve base capabilities nearly on par with the Transformer. Furthermore, we extend this validation to the more practical Top-1 Chunk Selection architecture — a direct generalization of the Top-1 Element Selection architecture, and implement GPU Kernels to ensure its time efficiency. Experiments show that the Top-1 Chunk Selection architecture, which adheres to our design principle, outperforms stateful architectures in base capabilities for both short (2k) and long (100k) sequences while maintaining competitive time efficiency.

The main contributions of our work are as follows:

- We propose a more indicative limited domain pre-training and out-of-distribution testing framework, successfully revealing the base capability degradation in stateful sequence modeling architectures during early pre-training stages. (Section 2)

- We investigate the impact of various common sequence modeling architecture factors on base capabilities and summarize the principle that "full-sequence arbitrary selection capability" is critical to avoiding degradation in base capabilities. (Section 3)

- We validate the principle using extremely simple Top-1 Element Selection architecture and demonstrate its generalization to the more practical Top-1 Chunk Selection architecture, accompanied by open-source GPU Kernels[1] to ensure time efficiency. (Section 4 and 5)

- Through our analysis and experiments, we prove the effectiveness of the proposed architecture design principle, providing valuable insights for future architecture designs.

## 2 Base Capabilities Evaluation

This work focuses on how sequence modeling architectures influence the base capabilities of pre-trained language models, thus necessitating the design of an evaluation framework and the implementation of assessments for different sequence modeling architectures.

### 2.1 Evaluation Scheme for Architecture Analysis

Existing sequence modeling architecture design works typically adopt the same **Mixed Domain Pre-Training** setting as large language model development, where corpora from as many domains as possible are collected and mixed as pre-training data. While this setting benefits practical large language model applications, it is detrimental to architecture analysis. It turns base capability tests

---

[1]https://github.com/luxinxyz/TSA

like language modeling into in-distribution evaluations, failing to reveal differences in architecture base capabilities during early pre-training stages. Moreover, it poorly predicts the true usability of different architectures when applied to unknown out-of-distribution domain tasks.

We tested this setting. Specifically, we pretrained models by mixing all domains (cc, c4, arxiv, book, github, stack and wiki) from the SlimPajama dataset [42] and retained a mixed-domain test set for evaluation. For models with ≈ 110M parameters, we pretrained Transformer [6], Transformer++ (with Rotary Embedding [43], GeGLU [41] and RMSNorm [56]), Transformer++ (Window=256), Based [2], Mamba-1 [16], Mamba-2 [11], RWKV-5 [32], RWKV-6 [32], RWKV-7 [33], DeltaNet [40] and Gated DeltaNet [51], with a sequence length of 2K and 100B tokens of pre-training. The scatter plots of pre-training loss versus mixed-domain test loss are shown in Figure 1(a). For models with ≈ 1.3B parameters, we pretrained Transformer++, Transformer++ (Window=256), Mamba-1 [16] and Mamba-2 [11] under similar settings, with results plotted in Figure 2(a).

From Figure 1(a) and Figure 2(a), it is evident that under the Mixed Domain Pre-Training setting, different sequence modeling architectures achieve similar test performance in language modeling when reaching comparable pre-training performance levels. Thus, this setting fails to reveal base capabilities difference among architectures during early pre-training and is unsuitable for architecture design and analysis.

To address this issue, we propose a **Limited Domain Pre-Training** with out-of-distribution (OOD) testing framework, where models are pretrained on a restricted set of domains and evaluated on unseen domains.

Since the SlimPajama dataset has already undergone deduplication across domain subsets, we easily tested this setting. Specifically, we pretrained models on the cc and c4 domains of SlimPajama and evaluated them on the arxiv, github and stack domains. Other pre-training settings remained similar, and the scatter plots of pre-training loss versus OOD test loss are shown in Figures 1(b) and 2(b).

From Figure 1(b) and Figure 2(b), significant differences emerge among sequence modeling architectures. At the same pre-training performance level, their OOD test performance varies substantially, confirming that base capabilities difference exist across architectures.

Additionally, we evaluated models with ≈ 1.3B parameter using the commonly adopted few-shot learning evaluation in prior work, which results in Figure 2(c). Similarly, no significant performance gaps were observed, suggesting that this evaluation method is also suboptimal for architecture design and analysis.

Based on these findings, we establish **Limited Domain Pre-Training** with OOD testing as our evaluation framework for base capabilities assessment. All subsequent tests in this work follow it.

## 2.2 Architecture-Induced Degradation of Base Capabilities

We further analyzed Figure 1(b) and Figure 2(b). The results show that only the standard attention-based Transformer and Transformer++ achieve optimal base capabilities—their OOD test performance is consistently the best at the same pre-training level, with no significant difference between them. In contrast, stateful sequence modeling architectures (e.g., Mamba, RWKV) exhibit varying degrees of base capabilities degradation, performing significantly worse than standard attention-based models under identical pre-training conditions.

These findings suggest that sequence modeling architectures directly induce base capabilities degradation, independent of data or other factors. The standard self-attention mechanism likely contains key architecture factors critical to base capabilities, some of which may be missing in stateful sequence modeling architectures.

## 3 Sequence Modeling Architecture Analysis

Based on the previous results, we know that the sequence modeling architecture can directly determine the model's base capabilities, and certain key architecture design factors are likely to play a significant role in these base capabilities. Therefore, identifying which architecture design factors are truly critical could be of great importance for subsequent architecture improvements or the design of new architectures.

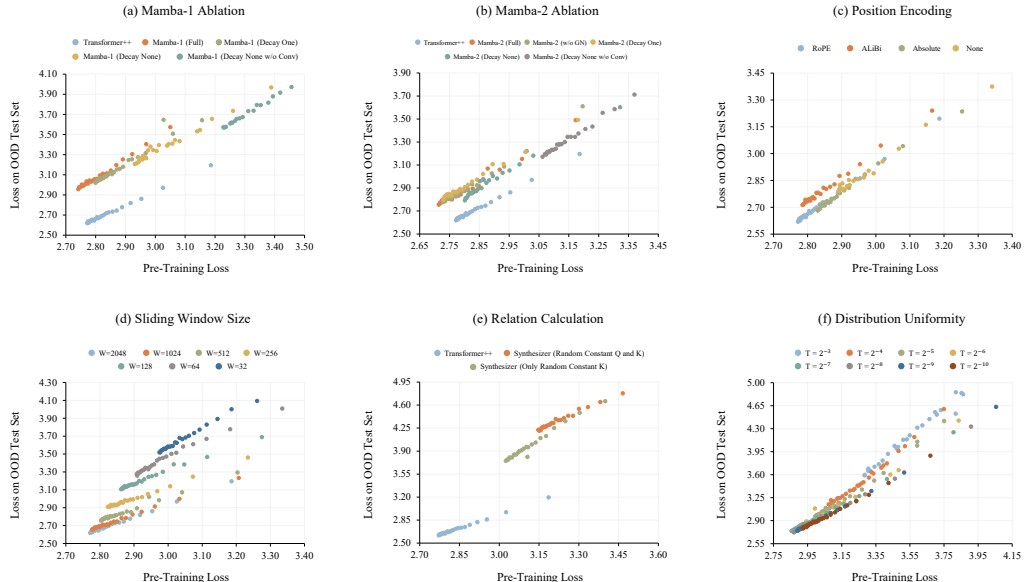

Figure 3: Analysis of the influence of various sequence modeling architecture components on base capabilities. (Model parameters≈110M, pre-trained tokens=100B or 25B and sequence length=2k)

## 3.1 Non-Determinative Factors of Base Capabilities

### 3.1.1 Mamba Ablation

Since the classical stateful sequence modeling architecture, Mamba, has been observed to exhibit degradation in base capabilities, we conducted ablation studies on some of its key components to assess their actual impact on base capabilities.

First, we focused on the **Data-Dependent Decay** in Mamba. By replacing the multi-dimensional independent data-dependent decay (Full) in Mamba-1 and Mamba-2 with either a shared data-dependent decay (Decay One) or no explicit decay (Decay None), we evaluated the effect of data-dependent decay on base capabilities, as shown in Figures 3(a) and 3(b). The results indicate that data-dependent decay only accelerates pre-training convergence but does not positively impact base capabilities. Specifically, at the same pre-training level, the out-of-distribution test performance of the ablated models did not decrease and even improved slightly.

Next, we examined the **Convolution** in Mamba. We further removed the convolution from the Decay None architecture (Decay None w/o Conv) to assess its effect on base capabilities, as shown in Figures 3(a) and 3(b). The results show that while convolution significantly speeds up pre-training convergence, it still does not contribute positively to base capabilities.

Finally, we investigated the **GroupNorm** in Mamba-2. By removing GroupNorm from the full architecture (Full → w/o GN), we evaluated its impact on base capabilities, as shown in Figure 3(b). The results confirm that GroupNorm also does not enhance base capabilities.

### 3.1.2 Position Encoding

Previous results have shown that Transformer and Transformer++ exhibit nearly identical base capabilities, with their primary difference in sequence modeling lying in their position encoding schemes. Therefore, we analyzed the impact of position encoding on base capabilities.

Specifically, we tested four common position encoding schemes: no position encoding, absolute position embedding, AliBi[34] and rotary position embedding [43], as illustrated in Figure 3(c). The results indicate that no position encoding, absolute position embedding and rotary position embedding yield very similar base capabilities, differing only in convergence speed. In contrast, AliBi exhibits some degradation in base capabilities. Based on these findings, we conclude that position encoding does not positively influence base capabilities.

### 3.2 Determinative Factors of Base Capabilities

#### 3.2.1 Full-Sequence Visibility

The first factor we identified as critical to base capabilities is sliding window size. This observation was prompted by the significant difference in base capabilities between Transformer++ and Transformer++ (Window=256), suggesting that sliding window size may be a key factor.

To investigate, we tested various sliding window sizes, as shown in Figure 3(d). The results demonstrate that as the sliding window size increases, both pre-training convergence speed and base capabilities improve significantly. Conversely, reducing the sequence context leads to gradual degradation in base capabilities. Thus, we conclude that **Full-Sequence Visibility** is an essential requirement for sequence modeling architectures.

#### 3.2.2 Real Relation Calculation

In previous results, we observed that under mixed domain pre-training settings, different models exhibited nearly identical test performance, creating the illusion that base capabilities is architecture-independent. This reminded us of the Synthesizer [45] series of models, where one variant replaced the true query-key computed scores with trainable random constant scores yet did not exhibit significant degradation in language modeling tasks. We hypothesized that this might also be due to mixed domain pre-training and that real relation computation between queries and keys could be a determinant of base capabilities.

To test this, we conducted experiments where we replaced keys with trainable random constants or replaced both queries and keys with trainable random constants, as shown in Figure 3(e). The results reveal that models without real relation computation suffer substantial degradation in base capabilities. Therefore, we conclude that **Real Relation Computation** is a necessary feature for sequence modeling architectures.

#### 3.2.3 Non-Uniform Distribution

Another key factor we examined is the uniformity of attention distribution in self-attention mechanisms. We reasoned that if attention distribution were entirely uniform, the model would degenerate into a naive averaging structure over values, which exhibits poor base capabilities. However, if we start from this structure and gradually introduce non-uniformity, the model would evolve toward normal attention distribution, which has strong base capabilities. This led us to suspect that distribution uniformity directly impacts base capabilities.

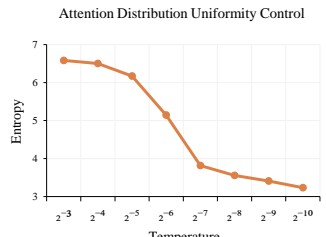

Figure 4: The relationship between attention distribution entropy and temperature.

To control attention distribution uniformity, we employed two techniques: 1) Adjusting distribution uniformity via the temperature of the Softmax function. 2) Applying normalization to queries and keys to ensure consistent temperature interpretation, unaffected by the optimization of query and key transformation parameters. We tested a series of models with varying temperatures and plotted the relationship between temperature and attention distribution entropy in Figure 4. The results show that as temperature decreases, attention distribution entropy also decreases, confirming that the distribution becomes more non-uniform.

Subsequently, we evaluated the out-of-distribution performance of these models, as shown in Figure 3(f). The results indicate that as temperature decreases (i.e., distribution becomes more non-uniform), base capabilities improves. Thus, we conclude that **Non-Uniform Distribution** is a necessary feature for sequence modeling architectures.

### 3.3 Key Architecture Design Principle: Full-Sequence Arbitrary Selection

Integrating the three key elements derived from the preceding analysis: Full-Sequence Visibility, Real Relation Calculation and Non-Uniform Distribution, we summarize them into a unified expression:

(a) Top-1 Element Selection Architecture (b) Top-1 Chunk Selection Kernel Design

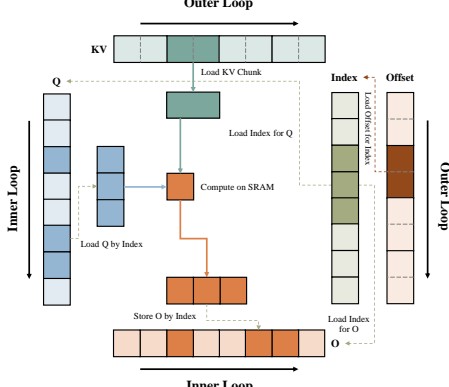

Figure 5: The Illustration include: (a) The overall architecture design of the Top-1 Element Selection architecture. (b) The kernel design for key component of the Top-1 Chunk Selection architecture.

*Supporting full-sequence arbitrary selection in sequence modeling architecture is the key architecture design principle for preventing the degradation of base capabilities.*

This principle directly impacts the model's base capabilities, and violating it will lead to degradation.

**Note that this is an architecture design principle rather than a capability principle.** The latter is often expressed as the model's retrieval capability. However, retrieval capability itself is not inherently tied to architecture design principles. For instance, stateful sequence modeling architectures like Mamba exhibit certain retrieval capabilities, yet they lack the architecture design of full-sequence arbitrary selection. It is this aspect of architecture design that our work truly focuses on.

## 4  Top-1 Element Selection Architecture

In previous section, we proposed a key architecture design principle for avoiding degradation in base capabilities. However, this conclusion was derived inductively and has not yet been experimentally validated. To address this, we designed an extremely minimalist **Top-1 Element Selection** architecture, which directly adheres to this design principle while maintaining strong base capabilities.

### 4.1  Architecture Design

We posit that "sequence modeling architectures with full-sequence arbitrary selection" is a key architecture design principle for preventing degradation in base capabilities. The simplest implementation of this principle is to directly select the element with the highest probability in the attention distribution as the output, as illustrated in Figure 5(a). We refer to this architecture as the **Top-1 Element Selection** architecture. In practice, it involves two additional operations: 1) applying normalization to queries and keys to enhance stability. 2) during training, attention scores are still computed, but the straight-through trick is introduced to reconcile top-1 selection with gradient updates.

As shown in Figure 5(a), the Top-1 Element Selection architecture is remarkably simple yet satisfies the design principle. It also incorporates three key elements: full-sequence visibility, real relation calculation and non-uniform distribution, making it an excellent candidate for validating our analysis.

### 4.2  Out-of-Distribution Generalization Evaluation

We pre-trained models with $\approx$ 110M and 1.3B parameters, maintaining the same pre-training settings as in previous experiments, and evaluated their OOD performance. The results, shown in Figures 6(a) and 6(b), demonstrate that the Top-1 Element Selection architecture achieves OOD performance

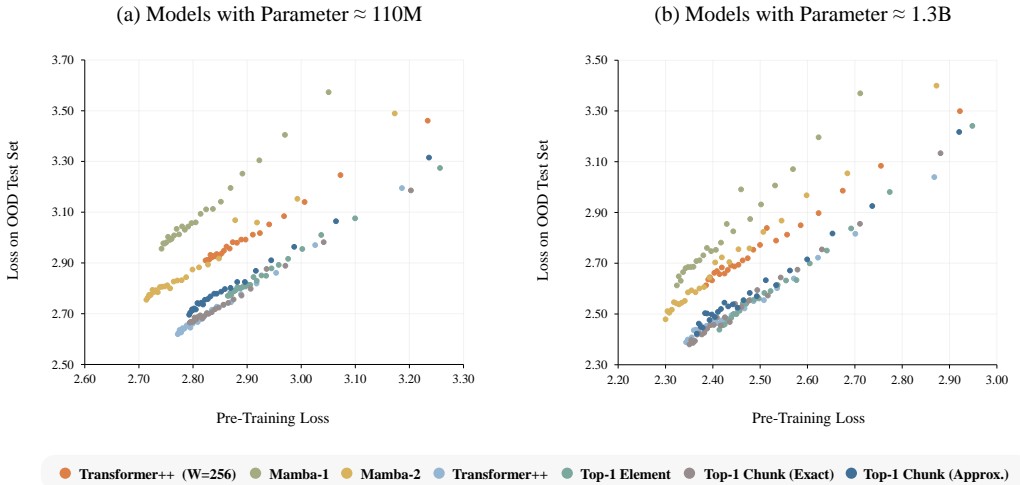

(a) Models with Parameter ≈ 110M    (b) Models with Parameter ≈ 1.3B

● Transformer++ (W=256)   ● Mamba-1   ● Mamba-2   ● Transformer++   ● Top-1 Element   ● Top-1 Chunk (Exact)   ● Top-1 Chunk (Approx.)

Figure 6: The out-of-distribution language modeling results of baselines and our Top-1 Element / Chunk Selection architectures. (Pre-trained tokens=100B, sequence length=2k and chunk size=128)

nearly on par with Transformer++ at both scales. This confirms that the architecture does not suffer from base capability degradation and validates the correctness of our proposed design principle.

## 4.3   Few-Shot Learning Evaluation

Although we previously argued that few-shot learning evaluation is not an ideal method for comparing base capabilities across architectures, it remains necessary to verify that the new architecture does not exhibit significant degradation in few-shot learning performance. Thus, we evaluated the few-shot learning performance of the 1.3B parameter model, as detailed in Table 2. The results show that the Top-1 Element Selection architecture achieves few-shot learning performance comparable to models like Transformer++, Mamba-1 and Mamba-2, indicating its strong performance in this setting.

Beyond standard tasks, we also evaluated retrieval tasks, a common benchmark in prior work. As shown in Table 1, the Top-1 Element Selection architecture significantly outperforms stateful sequence modeling architectures like Mamba-1 and Mamba-2 in retrieval tasks, further demonstrating its effectiveness.

| Model | SWDE (acc↑) | SQuAD (acc↑) | FDA (acc↑) |
|---|---|---|---|
| *Original Sequence Modeling Architecture* | | | |
| Transformer++ | **75.07** | **45.54** | **30.22** |
| *Stateful Sequence Modeling Architecture* | | | |
| Mamba-1 | 36.09 | 31.03 | 6.44 |
| Mamba-2 | 37.98 | 35.25 | 17.60 |
| Transformer++ (W=256) | 12.96 | 41.72 | 1.63 |
| *Architecture Based on the Analyzed Principles* | | | |
| Top-1 Element | 68.23 | 43.77 | 26.95 |
| Top-1 Chunk (Exact) | 69.22 | 44.40 | 22.32 |
| Top-1 Chunk (Approx.) | 61.66 | 45.38 | 27.13 |

Table 1: Results of retrieval tasks. (Model parameters ≈ 1.3B, pre-trained tokens= 100B, seq len=2k and chunk size=128)

## 5   Top-1 Chunk Selection Architecture

In the previous section, we designed the Top-1 Element Selection architecture, successfully validating our proposed architecture design principles. However, this architecture serves as a proof-of-concept and lacks practical utility. To address this, we extended it to the **Top-1 Chunk Selection** architecture and implemented GPU kernels to optimize efficiency while maintaining strong base capabilities.

### 5.1   Architecture Design

#### 5.1.1   Basic Design

The Top-1 Element Selection architecture has been verified to possess strong base capabilities but performs poorly in terms of pre-training convergence and time efficiency. To resolve this, we

| Model | MMLU (acc↑) | OBQA (acc_n↑) | ARC-e (acc_n↑) | ARC-c (acc_n↑) | BoolQ (acc↑) | RACE (acc↑) | SIQA (acc↑) | SCIQ (acc_n↑) | Hella. (acc_n↑) | COPA (acc↑) | PIQA (acc_n↑) | Wino. (acc↑) | WSC (acc↑) | Avg. Score |
|---|---|---|---|---|---|---|---|---|---|---|---|---|---|---|
| *Original Sequence Modeling Architecture* | | | | | | | | | | | | | | |
| Transformer++ | 25.21 | 33.60 | 48.48 | 25.94 | **60.43** | **33.88** | 39.56 | **77.60** | 48.42 | 74.00 | 68.99 | 53.91 | 65.57 | 50.43 |
| *Stateful Sequence Modeling Architecture* | | | | | | | | | | | | | | |
| Mamba-1 | 24.30 | **34.60** | **52.82** | 27.22 | 54.65 | 32.73 | 39.10 | 75.90 | 50.53 | 74.00 | 70.67 | 53.91 | 57.51 | 49.84 |
| Mamba-2 | 24.74 | 32.80 | 50.04 | **28.92** | 56.27 | 33.68 | 40.28 | 75.70 | **51.45** | **78.00** | **71.44** | **55.09** | 60.44 | 50.68 |
| Transformer++ (W=256) | 25.54 | 34.00 | 48.82 | 26.96 | 59.14 | **33.88** | 40.02 | 73.70 | 48.19 | 72.00 | 70.89 | 52.49 | 63.00 | 49.89 |
| *Architecture Based on the Analyzed Principles* | | | | | | | | | | | | | | |
| Top-1 Element | 25.94 | 32.60 | 47.85 | 26.45 | 60.15 | 31.29 | **40.94** | 73.40 | 45.05 | 72.00 | 69.15 | 53.75 | 62.27 | 49.30 |
| Top-1 Chunk (Exact) | **25.96** | 32.80 | 50.55 | 27.22 | **60.43** | 31.77 | 40.07 | 77.40 | 48.75 | 75.00 | 69.64 | 53.51 | **66.30** | **50.72** |
| Top-1 Chunk (Approx.) | 25.24 | 33.00 | 49.33 | 26.02 | 58.87 | 33.30 | 39.82 | 75.30 | 48.59 | 70.00 | 69.70 | 55.01 | 64.47 | 49.90 |

Table 2: The few-shot learning experimental results, where MMLU is 5-shot and the remaining tasks are 0-shot. (Model parameters≈1.3B, pre-trained tokens=100B, seq length=2k and chunk size=128)

generalized it to the Top-1 Chunk Selection architecture. The key difference from the Top-1 Element Selection architecture is that the query selection target shifts from fine-grained kv elements to coarse-grained kv chunks, and the attention mechanism operates only within the selected chunks. This preserves the design principles while enabling efficiency optimizations.

More specifically, the kv sequence is divided into multiple kv chunks. Each query attends to one selected full kv chunk (remote chunk) and the nearest partial kv chunk (local chunk), performing attention operations on them. Similar to the Top-1 Element Selection architecture, both queries and keys undergo normalization to enhance stability.

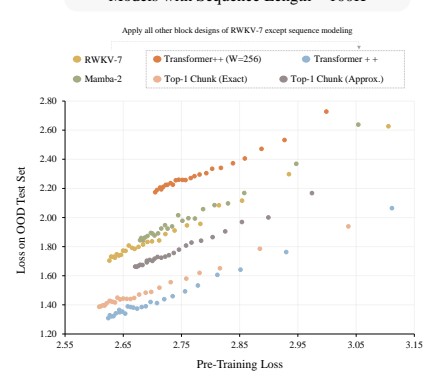

Figure 7: The OOD language modeling results of baselines and ours. (Model parameters ≈ 135M, pre-trained tokens=100B, seq length=100k and chunk size=128)

### 5.1.2 Exact and Approximate Variants

The selection of full kv chunks can be implemented in two ways, corresponding to two variants:

**Top-1 Chunk (Exact)**: The query computes the full attention distribution with all keys, obtaining the probability for each kv chunk. The chunk is selected based on these exact probabilities.

**Top-1 Chunk (Approx.)**: The exponential function in probability computation can be approximated by a first-order linear function. Under this approximation, the selection process simplifies to computing the mean vector of each key chunk and performing a dot product between the query and the mean vector to select the chunk. This significantly reduces computational overhead.

### 5.1.3 GPU Kernels Design

To achieve better time efficiency, we designed and implemented GPU kernels for the Top-1 Chunk Selection architecture. An overview is given in this section, with details in Appendix D .

For exact chunk selection, approximate chunk selection and local chunk attention, we directly implemented Triton kernels using simple, naive designs that already deliver good performance.

For remote chunk attention, we conducted specialized optimizations when implementing Triton Kernels, shown in Figure 5(b). Specifically, parallel processing is performed along the kv dimension in complete chunks, meaning different kv chunks are processed independently in parallel. This partitioning is feasible because each query selects only one remote chunk, ensuring no overlap between queries associated with different kv chunks. Within each kv chunk, we load index via offset, then load the queries relevant to the current kv chunk by the index and compute attention in SRAM, storing results by the index. Efficient offset and index handling is achieved by custom CUDA kernels.

During our research, we observed that DeepSeek and Kimi released two sparse attention works — NSA [53] and MoBA [27] in February 2025. Their sequence modeling approaches closely resemble our Top-1 Chunk (Approx.). **Although we discovered this architecture independently, we no longer claim Top-1 Chunk (Approx.) as our primary contribution.** Instead, our key contributions include Top-1 Element, Top-1 Chunk (Exact), analytical experiments and the released GPU kernels.

## 5.2 Out-of-Distribution Generalization and Few-Shot Learning Evaluation

We pre-trained models with $\approx$ 110M and 1.3B parameters (chunk size = 128), as shown in Figures 6(a) and 6(b). Results show both Top-1 Chunk Selection variants achieve OOD performance close to Transformer++ and outperform Mamba-1/2. We also evaluated few-shot learning tasks, as detailed in Tables 2 and 1. Top-1 Chunk Selection exhibits results and conclusions similar to Top-1 Element Selection: strong performance on general tasks without degradation and advantages in retrieval tasks.

## 5.3 Long Sequence and Architecture Combination

In this section, we evaluate two aspects: first, base capabilities, time efficiency and long-sequence retrieval capability of Top-1 Chunk Selection architecture on long sequences (100k); second, performance of Top-1 Chunk Selection architecture when transferred to other models. We designed an integrated setting where we ported Transformer++, Transformer++ (W=256), Top-1 Chunk (Exact) and Top-1 Chunk (Approx.) to the RWKV-7 architecture, replacing only the sequence modeling while retaining the rest of RWKV-7. Pre-training data remained unchanged, and OOD test data were filtered to retain samples from arxiv and github with original lengths exceeding 100k. Since stack contained almost no data with original lengths over 100k, we supplemented the test set with samples from AutoMathText [57] met this criterion.

| Model | Pre-Train Speed | Inference Speed | S-NIAH (Passkey Retrieval) | | | | |
|---|---|---|---|---|---|---|---|
| | | | 16k | 32k | 64k | 96k | 100k |
| *Original Architecture (Unmodified)* | | | | | | | |
| Mamba-2 | 5.57× | 1.01× | 5.40 | 1.40 | 0.40 | 0.20 | 0.80 |
| RWKV-7 | 2.47× | 0.73× | 70.80 | 14.80 | 1.20 | 0.00 | 0.20 |
| *Apply all other block designs of RWKV-7 except sequence modeling* | | | | | | | |
| Transformer++ | 1.00× | 1.00× | 76.20 | **49.80** | **25.60** | 16.20 | **15.80** |
| Transformer++ (W=256) | 5.29× | 1.72× | 1.60 | 0.40 | 0.20 | 0.20 | 0.40 |
| Top-1 Chunk (Exact) | 1.90× | 1.19× | 76.80 | **49.80** | 22.00 | 15.00 | 11.80 |
| Top-1 Chunk (Approx.) | 4.50× | 1.21× | **78.80** | 48.60 | 21.40 | **21.80** | 15.00 |

Table 3: Results of time efficiency and long sequence retrieval. (Model parameter$\approx$135M, pre-trained tokens=100B, seq length=100k and chunk size=128)

The base capabilities results are shown in Figure 7. Transformer++ demonstrated the strongest base capabilities, followed closely by Top-1 Chunk (Exact). Top-1 Chunk (Approx.) outperformed other stateful architectures in base capabilities, but due to its approximate nature, it exhibited significant pretraining convergence degradation and base capabilities degradation compared to Transformer++ (similar issues may also affect DeepSeek NSA and Kimi MoBA, which employ analogous approaches).

Time efficiency and long-sequence retrieval results are presented in Table 3. Top-1 Chunk (Approx.) achieved substantial speed improvements over Transformer++, reaching time efficiency levels comparable to Mamba-2. Top-1 Chunk (Exact) also showed speed gains, achieving time efficiency similar to RWKV-7. In long-sequence retrieval, results on S-NIAH [20] shown Top-1 Chunk Selection delivered performance close to Transformer++, while other stateful models suffered severe degradation.

# 6 Limitations

This work primarily focuses on models pre-trained with language modeling objectives, without exploring other pre-training objectives. As a result, our conclusions are limited to language models, narrowing the current applicability of our findings. We plan to expand this research with additional experiments in the future.

# 7 Conclusion

This work investigates how sequence modeling architectures influence base capabilities of pre-trained language models. We first reveal the degradation of base capabilities in stateful sequence modeling architectures by limited domain pre-training. Subsequently, via architecture analysis, we identify "full-sequence arbitrary selection" as the key architecture design principle for preventing such degradation. Finally, we validate our analysis by proposing Top-1 Element Selection and Top-1 Chunk Selection architecture, which may provide valuable references and foundations for future research.

## Acknowledgments

This work was supported by the New Generation Artificial Intelligence-National Science and Technology Major Project 2023ZD0121100, the National Natural Science Foundation of China (NSFC) via grant 62441614 and 62176078.

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

# A    Model Specifications and Pre-training Procedures

This work implemented and pre-trained multiple models with different architectures under various configurations to support experiments in different sections.

The first configuration involves models with approximately 110M (small scale) and 1.3B (large scale, available only for certain architectures) parameters, with a sequence length of 2k. It includes the following architectures: Transformer [6], Transformer++, Transformer++ (W=256), Based [2], Mamba-1 [16], Mamba-2 [11], RWKV-5 [32], RWKV-6 [32], RWKV-7 [33], DeltaNet [40], Gated DeltaNet [51], Top-1 Element, Top-1 Chunk (Exact) and Top-1 Chunk (Approx.).

Transformer adopts the GPT [35] architecture, is based on FlashAttention-2[2] [10], and has a hidden dimension of 768, 12 layers and 12 attention heads for the small-scale version; while the large-scale version has a hidden dimension of 2048, 24 layers and 16 attention heads. Transformer++ introduces Rotary Embedding [43], GeGLU [41] and RMSNorm [56] over the Transformer, also built using FlashAttention-2, with identical model specifications as the Transformer. Transformer++ (W=256) incorporates a window size of 256, retains the same model specification, and is also based on FlashAttention-2.

Based [2] is implemented according to the official open-source code[3], with a hidden dimension of 768 and 15 layers for the small-scale version; among them, there are 3 layers of window attention with 12 attention heads and a window size of 128, and 3 layers of TaylorExp linear attention with 12 attention heads. Mamba-1 [16] is implemented based on the official open-source code[4], with a hidden dimension of 768, 24 layers and a state size of 16 for the small-scale version; the large-scale version has a hidden dimension of 2048, 48 layers and the same state size of 16. Mamba-2 [11] is implemented based on the official open-source code (URL is the same as Mamba-1), with a hidden dimension of 768, 24 layers and a state size of 128 for the small-scale version; the large-scale version has a hidden dimension of 2048, 48 layers and the same state size of 128. RWKV-5/6/7 [32, 33] are implemented based on the official open-source code[5], with a hidden dimension of 768, 12 layers and 12 attention heads for the small-scale versions. DeltaNet [40] is based on Flash Linear Attention[6] [52], with a hidden dimension of 768, 12 layers and 8 attention heads for the small-scale version. Gated DeltaNet [51] is also based on Flash Linear Attention, with a hidden dimension of 768, 12 layers, 6 attention heads and a head dimension of 96 for the small-scale version.

Top-1 Element is implemented using PyTorch[7] and the transformers[8] library, matching the model specifications of the Transformer++. Top-1 Chunk (Exact/Approx.) shares the same model specifications as Transformer++, with a chunk size of 128. Its sequence modeling part is implemented using Triton[9] [46], and its offset and index CUDA kernel is modified from the fastmoe[10] [18].

In addition, all of the above models use the Mixtral [22] vocabulary, which contains 32,000 tokens. Large-scale models involved in the training process were trained using the DeepSpeed[11] open-source project. All models involving Rotary Embedding have a base of 10,000.

The second configuration involves models with approximately 135M parameters (small scale) and a sequence length of 100k. This mainly consists of architectures obtained by replacing the sequence modeling in RWKV-7. Specific models include: Transformer++, Transformer++ (W=256), Top-1 Chunk (Exact) and Top-1 Chunk (Approx.).

Transformer++ is based on FlashAttention-2, while Transformer++ (W=256) uses FlexAttention[12] [14]. The implementations of Top-1 Chunk (Exact) and Top-1 Chunk (Approx.) follow those described in the previous configuration. All four models have a hidden dimension of 768, 12

---

[2]https://github.com/Dao-AILab/flash-attention
[3]https://github.com/HazyResearch/based
[4]https://github.com/state-spaces/mamba
[5]https://github.com/BlinkDL/RWKV-LM
[6]https://github.com/fla-org/flash-linear-attention
[7]https://pytorch.org/
[8]https://github.com/huggingface/transformers
[9]https://github.com/triton-lang/triton
[10]https://github.com/laekov/fastmoe
[11]https://github.com/deepspeedai/DeepSpeed
[12]https://pytorch.org/blog/flexattention/

layers and 12 attention heads, with a chunk size of 128 for the two Top-1 Chunk models. Additionally, RWKV-7 was implemented using Flash Linear Attention when the sequence length was set to 100k to improve time efficiency. Gradient checkpointing was applied to reduce GPU memory consumption across all models. All models use the Mixtral vocabulary with a size of 32,000 tokens. Models incorporating Rotary Embedding maintain a base of 1,000,000.

All models were pre-trained from scratch on the SlimPajama [42] dataset using language modeling objective across 100B tokens (mixed domain or limited domain). These settings apply to most of the main experiments and analysis experiments in this work (with the exception being the "distribution uniformity" analysis experiment, where some models were pre-trained on 25B tokens). More detailed test data partitions and other related settings are introduced in Section 2.1 and Section 5.3. Pre-trained models were trained using bfloat16 precision and the Adam optimizer. Small-scale models used a learning rate of 2e-4, while large-scale models used a learning rate of 5e-5. The optimizer's $\beta_1$ and $\beta_2$ values were set to 0.9 and 0.95, respectively. Learning rates for small-scale models were warmed up over the first 2,500 steps, while those for large-scale models were warmed up over the first 5,000 steps, followed by linear decay. The batch sizes for small-scale models were 256 (sequence length=2k) and 8 (sequence length=100k), while that for large-scale models was 128 (sequence length=2k). All models were trained in a distributed manner across 8 Nvidia Tesla A100 GPUs, totaling approximately 1,221 days of single-GPU equivalent training time.

## B  Evaluation Procedures

For language modeling evaluation, since all models in this work were pre-trained from scratch using the same vocabulary, we directly use the loss on the test set for evaluation.

For few-shot learning evaluation, general tasks include MMLU [19], OpenBookQA [29], ARC Easy & Challenge [9], BoolQ [8], RACE [24], SIQA [39], SCIQ [48], HellaSwag [55], COPA [37], PIQA [5], WinoGrande [38] and Winograd [25]; retrieval tasks include SWDE [26], SQuAD [36] and FDA [3]. Among these, MMLU uses a 5-shot setting, while all others are evaluated in a 0-shot setting. To ensure the stability and reproducibility of the evaluation results, we use the lm-evaluation-harness[13] [15] open-source evaluation framework for evaluation.

For time efficiency evaluation (Table 3), we compare the pre-training speed of different models with an input length of 100k tokens and a total batch size of 8. For inference speed comparison, we use an input length of 50k tokens and generate 5k tokens, with a total batch size of 64.

For long-sequence retrieval evaluation (Table 3), to ensure the stability and reproducibility of the results, we use the RULER[14] [20] open-source evaluation framework for evaluation.

## C  Related Work

Our work is related to research on multiple topics, primarily including stateful sequence modeling architectures, sparse attention architectures and the analysis of sequence modeling architectures.

Stateful sequence modeling architectures have gradually evolved to address the inefficiency of standard self-attention [47] in handling long sequences. The core of these architectures lies in linear RNNs [28] and linear attention [23], with key characteristics being their ability to support both parallel and efficient training while also being expressible in the form of iterative state updates. Additionally, they offer certain advantages in terms of time and space efficiency compared to standard self-attention. Subsequent works have continuously introduced new structures or mechanisms to enhance the expressive power of stateful sequence modeling architectures. Early works incorporated data-independent decays, such as S4 [17], RetNet [44], RWKV-5 [32], among others. Later works added data-dependent decays, including Mamba-1 [16], Mamba-2 [11], and RWKV-6 [32]. The most recent developments have introduced the delta rule, exemplified by Gated DeltaNet [51] and RWKV-7 [33]. Our work does not involve the development of new stateful sequence modeling architectures; instead, it primarily reveals existing deficiencies in base capabilities within established stateful sequence modeling architectures.

---

[13]https://github.com/EleutherAI/lm-evaluation-harness
[14]https://github.com/NVIDIA/RULER

Sparse attention architectures represent another approach to addressing the efficiency issues of standard self-attention. These works mainly achieve improved computational efficiency by reducing the number of terms involved in attention calculations. Early efforts were dominated by manually designed fixed sparsity patterns, including Sparse Transformer [7], Longformer [4], BigBird [54], LongNet [13], and others. Recent works, such as NSA [53] and MoBA [27], introduce dynamic block selection-based sparsification schemes that effectively balance computational efficiency and practical performance. The core of our work is to reveal how sequence modeling architectures influence base capabilities. Although the improved architecture we ultimately propose also adopts a dynamic block selection-based sparsification scheme, unlike previous works that focus on heuristic designs from an efficiency perspective and only evaluate in-domain performance, we begin our analysis from base capabilities, gradually building toward a well-founded proposal for dynamic block selection. We further discover unique advantages of such architectures in out-of-distribution performance across domains and find that architectures like NSA and MoBA, which employ approximate selection strategies, may lead to limitations in out-of-distribution generalization capability. In addition, as mentioned in Section 5, although we discovered this architecture independently, we no longer claim Top-1 Chunk (Approx.) as our primary contribution. Instead, our key contributions include Top-1 Element, Top-1 Chunk (Exact), analytical experiments and the released GPU kernels.

Works focusing on the analysis of sequence modeling architectures mainly concentrate on revealing specific capability deficits in stateful sequence modeling architectures, particularly in tasks such as retrieval [49], copy [21], associative recall [1], and dynamic programming [50]. These studies have successfully conducted experimental validations on synthetic datasets and synthetic tasks. Additionally, some of these works analyze theoretical differences in representational capabilities between RNNs and Transformers. While our work also focuses on potential defects in stateful sequence modeling architectures, we primarily investigate deficiencies in base capabilities rather than specific ones — for example, whether such deficiencies can already be manifested in language modeling tasks. Moreover, our work does not delve into comparing strengths and weaknesses in model representation or retrieval capabilities but instead pays more attention to the architecture design itself, aiming to uncover truly effective components through changes in base capabilities and offering insights into architecture design.

## D  Algorithm Details

To achieve better time efficiency, we designed and implemented GPU kernels for the Top-1 Chunk Selection architecture.

Specifically, we designed Triton kernels for Top-1 Chunk Selection architecture, with its forward pass detailed in Algorithm 1 and its backward pass outlined in Algorithm 2 . Beyond these two core algorithms, the implementation involves several other components. In both algorithms, $\mathbf{F}$ and $\mathbf{I}$ represent the offset and index, respectively. The offset $\mathbf{F}$ sequentially records the number of times each chunk is selected by querys, while the index $\mathbf{I}$ sequentially records the indices of all queries associated with each chunk. These can be obtained through a naive PyTorch implementation; we developed hardware-efficient CUDA kernels based on the logic of the naive implementation. Additionally, the chunk indices for both exact selection and approximate selection can also be acquired via a naive PyTorch implementation. Similarly, we developed hardware-efficient Triton kernels following the logic of the naive implementation. Additionally, we designed Triton kernels for the decoding stage, which can effectively accelerate the decoding process. The source code has been released at: `https://github.com/luxinxyz/TSA` .

**Algorithm 1** Top-1 Chunk Selection Forward Pass

---

**Require:** $\mathbf{Q}, \mathbf{K}, \mathbf{V} \in \mathbb{R}^{T \times d}$, $\mathbf{F} \in \mathbb{R}^{\lceil \frac{T}{B_c} \rceil}$, $\mathbf{I} \in \mathbb{R}^{T-B_c}$, chunk size $B_c$, block size $B_r$.

1: Divide $\mathbf{K}, \mathbf{V}$ into $T_c = \left\lceil \frac{T}{B_c} \right\rceil$ blocks $\mathbf{K}_1, \ldots, \mathbf{K}_{T_c}$ and $\mathbf{V}_1, \ldots, \mathbf{V}_{T_c}$ of size $B_c \times d$ each, divide $\mathbf{F}$ into $T_c = \left\lceil \frac{T}{B_c} \right\rceil$ scalars $f_1, \ldots, f_{T_c}$.

2: Initialize $\mathbf{O} = (0)_{T \times d} \in \mathbb{R}^{T \times d}, \mathbf{M} = (-\infty)_T \in \mathbb{R}^T, \mathbf{L} = (0)_T \in \mathbb{R}^T, \mathbf{R} = (0)_T \in \mathbb{R}^T$.

3: **for** $1 \le j \le T_c$ **do**

4:     Load $\mathbf{K}_j, \mathbf{V}_j, f_j$ from HBM to on-chip SRAM.

5:     **if** $j = 1$ **then**

6:         On chip, Initialize $f_{j-1} = 0$.

7:     **else**

8:         Load $f_{j-1}$ from HBM to on-chip SRAM.

9:     **end if**

10:    On chip, compute $M = f_j - f_{j-1}$.

11:    Divide $\mathbf{I}_{[f_{j-1},\ldots,f_j]}$ into $M_r = \left\lceil \frac{M}{B_r} \right\rceil$ blocks $\mathbf{I}_1, \ldots, \mathbf{I}_{M_r}$ of size $B_r$ each.

12:    **for** $1 \le i \le M_r$ **do**

13:        Load $\mathbf{I}_i$ from HBM to on-chip SRAM.

14:        Load $\mathbf{Q}_i \in \mathbb{R}^{B_r \times d}$ by index $\mathbf{I}_i \in \mathbb{R}^{B_r}$ from $\mathbf{Q}$ in HBM to on-chip SRAM.

15:        On chip, compute $\mathbf{S}_{ij} = \mathbf{Q}_i \mathbf{K}_j^T \in \mathbb{R}^{B_r \times B_c}$.

16:        On chip, compute $\tilde{m}_{ij} = \mathrm{rowmax}(\mathbf{S}_{ij}) \in \mathbb{R}^{B_r}$.

17:        On chip, compute $\tilde{\mathbf{P}}_{ij} = \exp(\mathbf{S}_{ij} - \tilde{m}_{ij}) \in \mathbb{R}^{B_r \times B_c}$.

18:        On chip, compute $\tilde{\ell}_{ij} = \mathrm{rowsum}(\tilde{\mathbf{P}}_{ij}) \in \mathbb{R}^{B_r}$.

19:        On chip, compute $\mathbf{O}_i = \tilde{\mathbf{P}}_{ij} \mathbf{V}_j \in \mathbb{R}^{B_r \times d}$.

20:        Write $\mathbf{O}_i \in \mathbb{R}^{B_r \times d}$ by index $\mathbf{I}_i \in \mathbb{R}^{B_r}$ to $\mathbf{O}$ in HBM.

21:        Write $\tilde{m}_{ij} \in \mathbb{R}^{B_r}$ by index $\mathbf{I}_i \in \mathbb{R}^{B_r}$ to $\mathbf{M}$ in HBM.

22:        Write $\tilde{\ell}_{ij} \in \mathbb{R}^{B_r}$ by index $\mathbf{I}_i \in \mathbb{R}^{B_r}$ to $\mathbf{L}$ in HBM.

23:    **end for**

24: **end for**

25: Divide $\mathbf{Q}, \mathbf{K}, \mathbf{V}, \mathbf{O}$ into $T_c = \left\lceil \frac{T}{B_c} \right\rceil$ blocks $\mathbf{Q}_1, \ldots, \mathbf{Q}_{T_c}$, $\mathbf{K}_1, \ldots, \mathbf{K}_{T_c}$, $\mathbf{V}_1, \ldots, \mathbf{V}_{T_c}$ and $\mathbf{O}_1, \ldots, \mathbf{O}_{T_c}$ of size $B_c \times d$ each.

26: Divide $\mathbf{M}, \mathbf{L}$ into $T_c = \left\lceil \frac{T}{B_c} \right\rceil$ blocks $m_1, \ldots, m_{T_c}$ and $\ell_1, \ldots, \ell_{T_c}$ of size $B_c$ each.

27: **for** $1 \le i \le T_c$ **do**

28:    Load $\mathbf{Q}_i, \mathbf{K}_i, \mathbf{V}_i, \mathbf{O}_i, m_i, \ell_i$ from HBM to on-chip SRAM.

29:    On chip, compute $\mathbf{S}_i = \mathbf{Q}_i \mathbf{K}_i^T \in \mathbb{R}^{B_c \times B_c}$.

30:    On chip, compute $m_i^{(T)} = max(m_i, \mathrm{rowmax}(\mathbf{S}_i)) \in \mathbb{R}^{B_c}$.

31:    On chip, compute $\tilde{\mathbf{P}}_i = \exp(\mathbf{S}_i - m_i^{(T)}) \in \mathbb{R}^{B_c \times B_c}$.

32:    On chip, compute $\ell_i^{(T)} = e^{m_i - m_i^{(T)}} \ell_i + \mathrm{rowsum}(\tilde{\mathbf{P}}_i) \in \mathbb{R}^{B_c}$.

33:    On chip, compute $\mathbf{O}_i^{(T)} = \mathrm{diag}(\ell_i^{(T)})^{-1}(\mathrm{diag}(e^{m_i - m_i^{(T)}})\mathbf{O}_i + \tilde{\mathbf{P}}_i \mathbf{V}_i) \in \mathbb{R}^{B_c \times d}$.

34:    On chip, compute $r_i^{(T)} = m_i^{(T)} + log(\ell_i^{(T)}) \in \mathbb{R}^{B_c}$.

35:    Write $\mathbf{O}_i^{(T)} \in \mathbb{R}^{B_c \times d}$ to $\mathbf{O}$ in HBM.

36:    Write $r_i^{(T)} \in \mathbb{R}^{B_c}$ to $\mathbf{R}$ in HBM.

37: **end for**

38: Return $\mathbf{O}, \mathbf{R}$.

---

**Algorithm 2** Top-1 Chunk Selection Backward Pass

---

**Require:** $\mathbf{Q}, \mathbf{K}, \mathbf{V}, \mathbf{O}, \mathbf{dO} \in \mathbb{R}^{T \times d}, \mathbf{R} \in \mathbb{R}^T, \mathbf{F} \in \mathbb{R}^{\lceil \frac{T}{B_c} \rceil}, \mathbf{I} \in \mathbb{R}^{T-B_c}$, chunk size $B_c$, block size $B_r$.

1: Divide $\mathbf{K}, \mathbf{V}$ into $T_c = \left\lceil \frac{T}{B_c} \right\rceil$ blocks $\mathbf{K}_1, \ldots, \mathbf{K}_{T_c}$ and $\mathbf{V}_1, \ldots, \mathbf{V}_{T_c}$ of size $B_c \times d$ each, divide
$\quad$ $\mathbf{F}$ into $T_c = \left\lceil \frac{T}{B_c} \right\rceil$ scalars $f_1, \ldots, f_{T_c}$.
2: Initialize $\mathbf{dQ} = (0)_{T \times d} \in \mathbb{R}^{T \times d}, \mathbf{dK} = (0)_{T \times d} \in \mathbb{R}^{T \times d}, \mathbf{dV} = (0)_{T \times d} \in \mathbb{R}^{T \times d}$.
3: **for** $1 \leq j \leq T_c$ **do**
4: $\quad$ Load $\mathbf{K}_j, \mathbf{V}_j, f_j$ from HBM to on-chip SRAM.
5: $\quad$ Initialize $\mathbf{dK}_j = (0)_{B_c \times d} \in \mathbb{R}^{B_c \times d}, \mathbf{dV}_j = (0)_{B_c \times d} \in \mathbb{R}^{B_c \times d}$.
6: $\quad$ **if** $j = 1$ **then**
7: $\quad\quad$ On chip, Initialize $f_{j-1} = 0$.
8: $\quad$ **else**
9: $\quad\quad$ Load $f_{j-1}$ from HBM to on-chip SRAM.
10: $\quad$ **end if**
11: $\quad$ On chip, compute $M = f_j - f_{j-1}$.
12: $\quad$ Divide $\mathbf{I}_{[f_{j-1}, \ldots, f_j]}$ into $M_r = \left\lceil \frac{M}{B_r} \right\rceil$ blocks $\mathbf{I}_1, \ldots, \mathbf{I}_{M_r}$ of size $B_r$ each.
13: $\quad$ **for** $1 \leq i \leq M_r$ **do**
14: $\quad\quad$ Load $\mathbf{I}_i$ from HBM to on-chip SRAM.
15: $\quad\quad$ Load $\mathbf{Q}_i \in \mathbb{R}^{B_r \times d}$ by index $\mathbf{I}_i \in \mathbb{R}^{B_r}$ from $\mathbf{Q}$ in HBM to on-chip SRAM.
16: $\quad\quad$ Load $\mathbf{O}_i \in \mathbb{R}^{B_r \times d}$ by index $\mathbf{I}_i \in \mathbb{R}^{B_r}$ from $\mathbf{O}$ in HBM to on-chip SRAM.
17: $\quad\quad$ Load $\mathbf{dO}_i \in \mathbb{R}^{B_r \times d}$ by index $\mathbf{I}_i \in \mathbb{R}^{B_r}$ from $\mathbf{dO}$ in HBM to on-chip SRAM.
18: $\quad\quad$ Load $r_i \in \mathbb{R}^{B_r}$ by index $\mathbf{I}_i \in \mathbb{R}^{B_r}$ from $\mathbf{R}$ in HBM to on-chip SRAM.
19: $\quad\quad$ On chip, compute $\mathbf{S}_{ij} = \mathbf{Q}_i \mathbf{K}_j^T \in \mathbb{R}^{B_r \times B_c}$.
20: $\quad\quad$ On chip, compute $\mathbf{P}_{ij} = \exp(\mathbf{S}_{ij} - r_i) \in \mathbb{R}^{B_r \times B_c}$ .
21: $\quad\quad$ On chip, compute $\mathbf{dV}_j \leftarrow \mathbf{dV}_j + \mathbf{P}_{ij}^\top \mathbf{dO}_i \in \mathbb{R}^{B_c \times d}$.
22: $\quad\quad$ On chip, compute $\mathbf{dP}_{ij} = \mathbf{dO}_i \mathbf{V}_j^\top \in \mathbb{R}^{B_r \times B_c}$.
23: $\quad\quad$ On chip, compute $\mathbf{D}_i = \mathrm{rowsum}(\mathbf{dO}_i \circ \mathbf{O}_i) \in \mathbb{R}^{B_r}$.
24: $\quad\quad$ On chip, compute $\mathbf{dS}_{ij} = \mathbf{P}_{ij} \circ (\mathbf{dP}_{ij} - \mathbf{D}_i) \in \mathbb{R}^{B_r \times B_c}$.
25: $\quad\quad$ On chip, compute $\mathbf{dK}_j \leftarrow \mathbf{dK}_j + \mathbf{dS}_{ij}^\top \mathbf{Q}_i \in \mathbb{R}^{B_c \times d}$.
26: $\quad\quad$ On chip, compute $\mathbf{dQ}_i = \mathbf{dS}_{ij} \mathbf{K}_j \in \mathbb{R}^{B_r \times d}$.
27: $\quad\quad$ Write $\mathbf{dQ}_i \in \mathbb{R}^{B_r \times d}$ by index $\mathbf{I}_i \in \mathbb{R}^{B_r}$ to $\mathbf{dQ}$ in HBM.
28: $\quad$ **end for**
29: $\quad$ Write $\mathbf{dK}_j \in \mathbb{R}^{B_c \times d}, \mathbf{dV}_j \in \mathbb{R}^{B_c \times d}$ to $\mathbf{dK}, \mathbf{dV}$ in HBM.
30: **end for**
31: Divide $\mathbf{Q}, \mathbf{K}, \mathbf{V}, \mathbf{O}$ into $T_c = \left\lceil \frac{T}{B_c} \right\rceil$ blocks $\mathbf{Q}_1, \ldots, \mathbf{Q}_{T_c}$, $\mathbf{K}_1, \ldots, \mathbf{K}_{T_c}$, $\mathbf{V}_1, \ldots, \mathbf{V}_{T_c}$ and
$\quad$ $\mathbf{O}_1, \ldots, \mathbf{O}_{T_c}$ of size $B_c \times d$ each.
32: Divide $\mathbf{dQ}, \mathbf{dK}, \mathbf{dV}, \mathbf{dO}$ into $T_c = \left\lceil \frac{T}{B_c} \right\rceil$ blocks $\mathbf{dQ}_1, \ldots, \mathbf{dQ}_{T_c}$, $\mathbf{dK}_1, \ldots, \mathbf{dK}_{T_c}$,
$\quad$ $\mathbf{dV}_1, \ldots, \mathbf{dV}_{T_c}$ and $\mathbf{dO}_1, \ldots, \mathbf{dO}_{T_c}$ of size $B_c \times d$ each.
33: Divide $\mathbf{R}$ into $T_c = \left\lceil \frac{T}{B_c} \right\rceil$ blocks $r_1, \ldots, r_{T_c}$ of size $B_c$ each.
34: **for** $1 \leq i \leq T_c$ **do**
35: $\quad$ Load $\mathbf{Q}_i, \mathbf{K}_i, \mathbf{V}_i, \mathbf{O}_i, \mathbf{dQ}_i, \mathbf{dK}_i, \mathbf{dV}_i, \mathbf{dO}_i, r_i$ from HBM to on-chip SRAM.
36: $\quad$ On chip, compute $\mathbf{S}_i = \mathbf{Q}_i \mathbf{K}_i^T \in \mathbb{R}^{B_c \times B_c}$.
37: $\quad$ On chip, compute $\mathbf{P}_i = \exp(\mathbf{S}_i - r_i) \in \mathbb{R}^{B_c \times B_c}$ .
38: $\quad$ On chip, compute $\mathbf{dV}_i \leftarrow \mathbf{dV}_i + \mathbf{P}_i^\top \mathbf{dO}_i \in \mathbb{R}^{B_c \times d}$.
39: $\quad$ On chip, compute $\mathbf{dP}_i = \mathbf{dO}_i \mathbf{V}_i^\top \in \mathbb{R}^{B_c \times B_c}$.
40: $\quad$ On chip, compute $\mathbf{D}_i = \mathrm{rowsum}(\mathbf{dO}_i \circ \mathbf{O}_i) \in \mathbb{R}^{B_c}$.
41: $\quad$ On chip, compute $\mathbf{dS}_i = \mathbf{P}_i \circ (\mathbf{dP}_i - \mathbf{D}_i) \in \mathbb{R}^{B_c \times B_c}$.
42: $\quad$ On chip, compute $\mathbf{dQ}_i \leftarrow \mathbf{dQ}_i + \mathbf{dS}_i \mathbf{K}_i \in \mathbb{R}^{B_c \times d}$.
43: $\quad$ On chip, compute $\mathbf{dK}_i \leftarrow \mathbf{dK}_i + \mathbf{dS}_i^\top \mathbf{Q}_i \in \mathbb{R}^{B_c \times d}$.
44: $\quad$ Write $\mathbf{dQ}_i \in \mathbb{R}^{B_c \times d}, \mathbf{dK}_i \in \mathbb{R}^{B_c \times d}, \mathbf{dV}_i \in \mathbb{R}^{B_c \times d}$ to $\mathbf{dQ}, \mathbf{dK}, \mathbf{dV}$ in HBM.
45: **end for**
46: Return $\mathbf{dQ}, \mathbf{dK}, \mathbf{dV}$.

---

# E    More Results Under Mixed Domain Pre-Training Setting

Although our primary focus is on experiments under the limited domain pre-training setting, we also conducted preliminary experiments on the Top-1 Element Selection Architecture under mixed domain pre-training setting during the early stages of this work. The results, as shown in Figures 8 and 9, indicate that the performance of the Top-1 Element Selection Architecture aligns with the conclusions drawn in the paper, with no degradation in base capabilities. This suggests that the usability of the proposed architecture remains consistent across different pre-training settings.

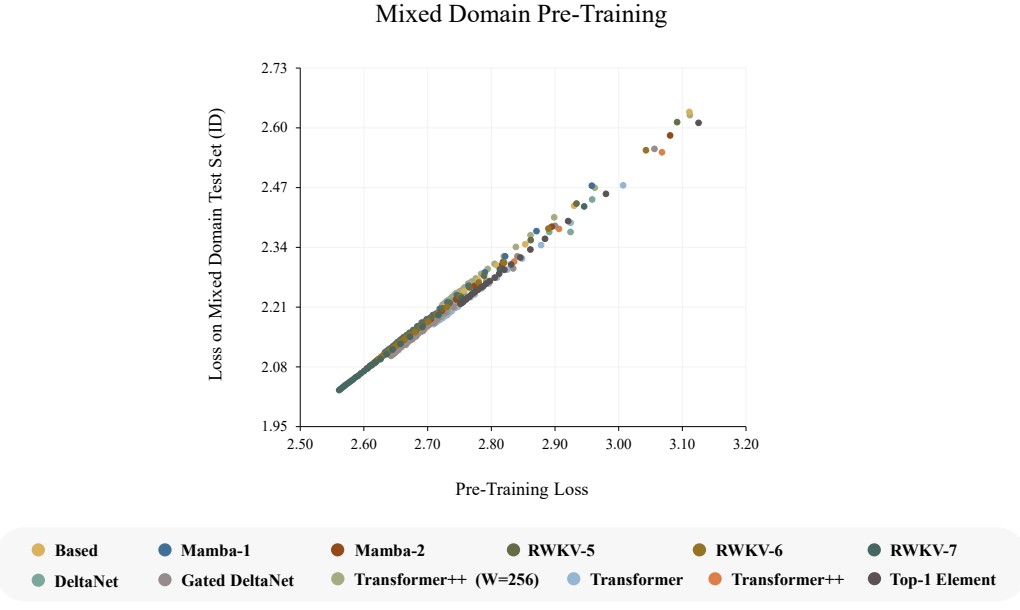

Figure 8: Language modeling results of various sequence modeling architectures under mixed domain pre-training settings. (Model parameters≈110M, pre-trained tokens=100B and sequence length=2k)

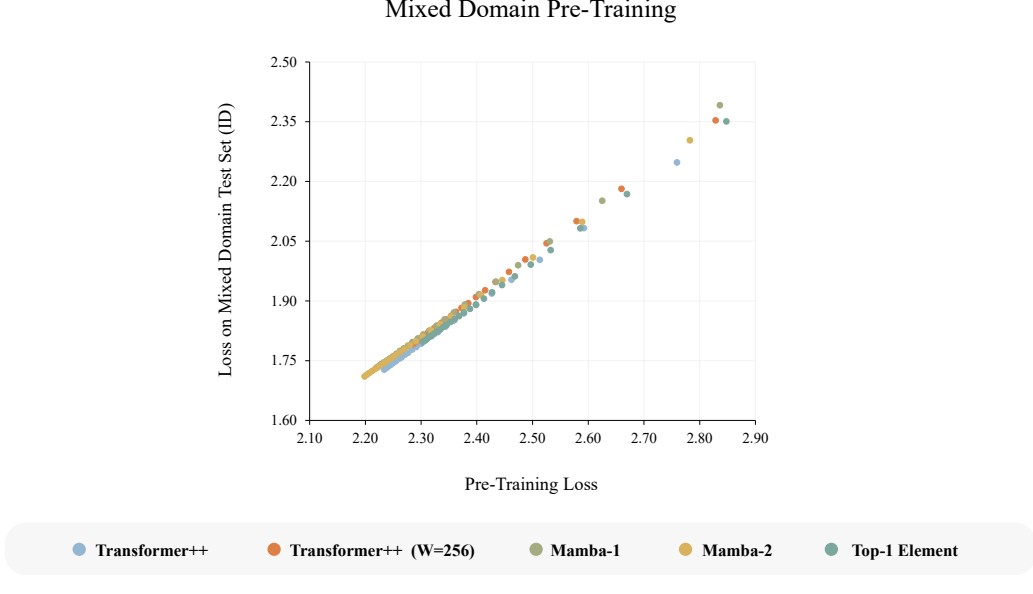

Figure 9: Language modeling results of various sequence modeling architectures under mixed domain pre-training settings. (Model parameters≈1.3B, pre-trained tokens=100B and sequence length=2k)

