# OpenReview forum: "How Does Sequence Modeling Architecture Influence Base Capabilities of Pre-trained Language Models? Exploring Key Architecture Design Principles to Avoid Base Capabilities Degradation"
_NeurIPS.cc/2025/Conference — NeurIPS 2025 poster_

### Official Review · Reviewer_z1L8 · 2025-06-16

**Clarity:** 3
**Significance:** 3
**Originality:** 3
**Rating:** 4
**Confidence:** 4

**Summary:**

This paper explores the design principles of sequence modeling architectures and their performance across various tasks, particularly in enhancing fundamental capabilities and handling long sequences effectively. The paper first introduces a key design principle: "sequence modeling architectures must possess the capability of arbitrary selection across the entire sequence," aimed at preventing the degradation of foundational abilities.

They then provide a detailed description of two methods for implementing full key-value (KV) block selection: the Exact and Approximate (Approx.) variants.  This simplifies the process of calculating the dot product between the average vector of each key block and the query, thereby significantly reducing computational overhead.

In addition, the paper discusses differences in model performance under various pre-training setups. The study finds that although mixed-domain pre-training is beneficial for practical applications of large-scale language models, it transforms fundamental capability testing into in-distribution evaluation. This approach fails to reveal differences in foundational capabilities among different architectures during the early pre-training stages.

Finally, the paper presents several experimental results showing that both Top-1 element selection and Top-1 block selection methods outperform state-of-the-art architectures in fundamental capabilities on both short sequences (2k) and long sequences (100k), while maintaining competitive time efficiency.

**Questions:**

See Weaknesses

**Ethical Concerns:**

["NO or VERY MINOR ethics concerns only"]

**Final Justification:**

Yes

**Limitations:**

Yes

**Paper Formatting Concerns:**

The rebuttal is clear. The authors address my concerns.

**Quality:**

4

**Strengths And Weaknesses:**

**Strengths**

+ Despite the unclear practical implications of OOD loss, especially since the authors avoid discussing its connection with retrieval capabilities, the paper provides a highly insightful perspective on revealing the differences in capabilities among various models and components. It identifies that supporting full-sequence arbitrary selection in sequence modeling architecture is the key design principle for preventing degradation of base capabilities.

+ The authors propose a toy architecture to validate their viewpoint and implement Triton and CUDA operators, achieving efficient training on contexts up to 100k in length with minimal performance loss.

**Weaknesses**

+ The exact **nature of what OOD loss or base capabilities reflect about a model's abilities remains unclear**. While the authors mention in the introduction that strong base capabilities can lead to robust language modeling and few-shot learning abilities, the results in Figure 2 show that OOD Loss and few-shot performance **do not seem strongly correlated**.

+ Further questions regarding base capabilities: **Does the author consider retrieval capabilities as part of base capabilities**? Different data types have varying impacts on different architectures. Linear attention models are generally thought to require additional context-dependent data to aid in learning retrieval capabilities. Would adding such data to both transformers and Mamba models reduce the performance gap between them?

+ Although this point does not affect the validity of the article, it raises concern: Typically, pre-training lengths are 4k or 8k, but the maximum length used by the authors when testing full-sequence visibility was 2k. It is uncertain whether conclusions drawn from longer sequences would remain valid.

+ Additionally, during the testing of full-sequence visibility, the **terminology "window size"** used by the authors could be misleading. Clarification is needed on whether it refers to the sample length during training or if a sliding window attention mechanism was employed.

---

> ### Author Rebuttal · Authors · 2025-07-31
>
> Thanks for your insightful review and valuable feedback!
>
> We answer your questions below.
>
> ---
>
> **Q1:** The exact nature of what OOD loss or base capabilities reflect about a model's abilities remains unclear. While the authors mention in the introduction that strong base capabilities can lead to robust language modeling and few-shot learning abilities, the results in Figure 2 show that OOD Loss and few-shot performance do not seem strongly correlated.
>
> **A1:** We appreciate your feedback and would like to clarify some potentially confusing expressions in our manuscript:
>
> In our work, **"base capabilities"** refers to an attribute of pre-trained models that influences their ultimate performance when adapted to downstream tasks.
>
> While base capabilities can be measured through various approaches, prior work on stateful sequence modeling primarily evaluated it via **in-distribution (ID) generalization capability of language modeling** or **few-shot learning capability**. However, we found these metrics insufficient for revealing architecture differences between sequence modeling approaches. Consequently, we proposed using **out-of-distribution (OOD) generalization capability of language modeling** as a more discriminative measurement, which led to several meaningful discoveries.
>
> Thus, your observation that **"the results in Figure 2 show that OOD Loss and few-shot performance do not seem strongly correlated" actually aligns with our expectations**. Precisely because few-shot performance proved inadequate for distinguishing base capabilities differences between architectures, we introduced OOD loss as an alternative metric. This doesn't reflect an intrinsic contradiction in base capabilities, but rather demonstrates that different measurement approaches reveal distinct levels of capability differences. **The OOD loss captures more profound architecture distinctions, which explains its weaker correlation with the metrics reflecting more superficial differences.**
>
> ---
>
> **Q2:** Further questions regarding base capabilities: Does the author consider retrieval capabilities as part of base capabilities? Different data types have varying impacts on different architectures. Linear attention models are generally thought to require additional context-dependent data to aid in learning retrieval capabilities. Would adding such data to both transformers and Mamba models reduce the performance gap between them?
>
> **A2:** Whether to consider **"retrieval capability"** as part of the base capabilities may be a matter of definition. **In our paper,** we did not define it as a base capability but rather treated it as a specialized skill, though defining it as base capability would not be incorrect.
>
> Our decision not to regard **retrieval capability** as being on par with **language modeling OOD generalization** as a base capability stems primarily from the observation that **it appears more easily satisfied than the latter**. The key evidence lies in our 100k sequence-length experiments, where Top-1 Chunk (Approx.) and Top-1 Chunk (Exact) demonstrated nearly identical retrieval performance (Table 3), yet Top-1 Chunk (Approx.) still showed significant degradation in language modeling OOD generalization (Figure 7). Therefore, we prefer using language modeling OOD generalization as our measure of base capabilities, which also has the advantage of being more closely aligned with the model's original pre-training objective.
>
> You raised an insightful point about **different data types affect architectures differently** - this indeed presents an alternative perspective where data could be treated as an architecture-dependent dynamic variable. However, **much current research still tends to view Transformer as the ideal reference point for capabilities, considering it a weakness when other efficient sequence modeling architectures require specific data to complete certain capabilities**. Our work follows this conventional approach, aiming to design architectures that match Transformer's data efficiency without capability degradation. Thus, regarding your question about whether adding retrieval-oriented auxiliary training data to both Transformer and Mamba would narrow their retrieval performance gap: while this might help, we would still interpret it as reflecting Mamba's inherent limitations. **That said**, we fully acknowledge that your perspective of treating data as an architecture-dependent dynamic variable is more profound, and we hope to explore this direction in future work.
>
> ---
>
> **Q3:** Although this point does not affect the validity of the article, it raises concern: Typically, pre-training lengths are 4k or 8k, but the maximum length used by the authors when testing full-sequence visibility was 2k. It is uncertain whether conclusions drawn from longer sequences would remain valid.
>
> **A3:** We understand your concerns and would like to provide the following clarification:
>
> The majority of our pre-training experiments were conducted with 2k sequences, including the relevant experiments that established the principle of full-sequence arbitrary selection. However, **we additionally performed pre-training experiments with 100k sequence lengths (Figure 7)**. The out-of-distribution experimental results remain consistent with the principles summarized from the 2k sequence experiments, with no violations observed. **We believe this supplementary experiment helps address the limitations of using only 2k sequence lengths to some extent.**
>
> ---
>
> **Q4:** Additionally, during the testing of full-sequence visibility, the terminology "window size" used by the authors could be misleading. Clarification is needed on whether it refers to the sample length during training or if a sliding window attention mechanism was employed.
>
> **A4:** We sincerely apologize for any confusion caused. Please allow us to clarify:
>
> The term **"window size"** used in testing full-sequence visibility refers to the **sliding window size**, while the sample length during training remains unchanged at 2k. We will revise the relevant expressions according to your comments to prevent any potential misunderstandings.
>
> ---

---

> > ### Comment · Reviewer_z1L8 · 2025-08-06
> >
> > Thank you for your response. I will increase my score.

---

### Official Review · Reviewer_EQyK · 2025-06-29

**Clarity:** 2
**Significance:** 2
**Originality:** 2
**Rating:** 4
**Confidence:** 4

**Summary:**

The paper studies the base capabilities of LLMs through OOD tests. And the paper discovers that full selection is a key property and designs the top-chunk selection architecture.

**Questions:**

See Weaknesses.

**Ethical Concerns:**

["NO or VERY MINOR ethics concerns only"]

**Final Justification:**

The rebuttal is clear. The authors justify my questions. But due to the novelty limitation, I only increase my score by 1.

**Limitations:**

Yes

**Quality:**

2

**Strengths And Weaknesses:**

Strengths:
1. The topic is an important topic: how to evaluate architecture designs besides the current pretraining method.
2. The discovery is interesting. Top-chunk attention can achieve similar performances as full attention.

Weaknesses:
1. It's unclear what the definition of base capabilities is. The author seems to evaluate it by OOD tests. What is base capabilities? Why can OOD test evaluate the base capabilities? These two questions need to be answered and emphasized clearly in the main part of the paper. Currently both are vauge.
2. The novelty is limited. As the authors mentioned themselves, MOBA and NSA already proposed top chunk selection architectures. The implementation is not novel enough.
3. The connection between full-selection and the top-element training seems to be not strong enough. From my understanding, full-selection is an inference concept. So the authors need to inference with only one element. But the authors seem to training only backward through certain chunk/element but inference, from my understanding, uses the full sequence.

---

> ### Author Rebuttal · Authors · 2025-07-31
>
> Thanks for your insightful review and valuable feedback!
>
> We answer your questions below.
>
> ---
>
> **Q1:** It's unclear what the definition of base capabilities is. The author seems to evaluate it by OOD tests. What is base capabilities? Why can OOD test evaluate the base capabilities? These two questions need to be answered and emphasized clearly in the main part of the paper. Currently both are vauge.
>
> **A1:** We sincerely apologize for any confusion caused and would like to provide the following clarification:
>
> Pre-trained models can achieve performance improvements in areas such as few-shot learning through training solely on language modeling tasks. **This suggests that pre-trained models possess certain base capabilities before being specifically adapted to downstream tasks.**
>
> Regarding base capabilities, there exist various measurement approaches. Previous work on stateful sequence modeling primarily measured them through **in-distribution (ID) generalization capability of language modeling** or **few-shot learning capability.** However, we found these metrics insufficient for effectively distinguishing between different sequence modeling architectures. Therefore, we proposed using **out-of-distribution (OOD) generalization capability of language modeling** as a measure of base capabilities, which proves to be a more discriminative approach and has led to several meaningful findings.
>
> **To summarize,** we employ OOD generalization of language modeling as our measure of base capabilities. We argue that this approach is valid because it inherits aspects of previous language modeling-based measurement methods while **offering better discriminative power for comparing different sequence modeling architectures.** However, as you pointed out, we did lack specific explanations about this in the earlier sections of our paper. We will carefully follow your suggestion to make corresponding improvements.
>
> ---
>
> **Q2:** The novelty is limited. As the authors mentioned themselves, MOBA and NSA already proposed top chunk selection architectures. The implementation is not novel enough.
>
> **A2:** As you mentioned, we acknowledge in the paper that our independently proposed **Top-1 Chunk (Approx.)** shares certain similarities with MOBA and NSA, and we explicitly state that due to the timing of submission, we do not position it as our primary contribution. The main contributions we claim in the paper include **Top-1 Element, Top-1 Chunk (Exact), architectural analysis conclusions, and released GPU kernels**, which better demonstrate the uniqueness of our work.
>
> Furthermore, regarding the Top-1 Chunk architecture, **we have made two additional contributions that go beyond the scope of MOBA and NSA:**
>
> 1. Rather than designing this architecture directly from an efficiency perspective, we derived it through step-by-step analysis of architecture factors affecting capability, thereby revealing its deeper advantages.
>
> 2. In Figure 7, we demonstrate that Top-1 Chunk still exhibits capability limitations due to approximate selection, which to some extent suggests that MOBA and NSA might share similar capability limitations. This provides an important starting point for future research.
>
> Therefore, **we believe our work still possesses significant unique aspects** that should compensate for the lack of primacy regarding the Top-1 Chunk selection architecture.
>
> ---
>
> **Q3:** The connection between full-selection and the top-element training seems to be not strong enough. From my understanding, full-selection is an inference concept. So the authors need to inference with only one element. But the authors seem to training only backward through certain chunk/element but inference, from my understanding, uses the full sequence.
>
> **A3:** We appreciate your feedback and would like to clarify some points that may have caused confusion:
>
> Regarding your mention of **"full-selection,"** we understand this likely refers to our concept of **"Full-Sequence Arbitrary Selection"** in the paper. We suspect this terminology may have been misleading. To clarify, this does not imply physically selecting all sequence elements for weighted summation. Rather, it denotes the architecture's inherent capability to freely select any individual element from the sequence. As implemented in our Top-1 Element architecture, this mechanism actually chooses only the element with the maximum dot-product while disregarding others (i.e., no weighted summation occurs).
>
> We should emphasize that the Top-1 Element architecture is not designed as a computationally efficient framework for either training or inference. It serves as a transitional architecture, with computational costs comparable to standard Multi-Head Attention, primarily for analysis purposes. **Although both training and inference involve selecting just one element from the full sequence, the architecture still requires maximum inner-product search across the full sequence.** Thus, the computations are performed over the entire sequence, only a single element is ultimately selected.
>
> These explanations are offered based on our understanding of your concerns. However, if we have misinterpreted your original intention, we would be grateful for any additional clarification you could provide!
>
> ---

---

> > ### Comment · Reviewer_EQyK · 2025-08-05
> > **Thanks for the response**
> >
> > Thanks for the response. The rebuttal is clear. I increased my score.

---

### Official Review · Reviewer_zo9Q · 2025-07-01

**Clarity:** 1
**Significance:** 3
**Originality:** 3
**Rating:** 5
**Confidence:** 4

**Summary:**

The authors study the language modeling performance of popular architectures, including Transformer++, Mamba, RWKV, and DeltaNet, on an out-of-distribution (OOD) held-out set compared to an in-distribution (ID) held-out set. While some architectures converge faster and achieve a lower ID loss in the same number of gradient steps, none surpass the Transformer's ability to generalize (as measured by the OOD performance relative to the ID one). The authors conducted ablation studies on Mamba's decay, convolution, normalization, and the Transformer’s positional encodings, revealing that these design choices impact convergence speed but not generalization. Instead, the authors identify three mechanisms as critical for generalization: full sequence visibility, direct token-to-token relationship computation, and a non-uniform attention distribution. To validate this, they evaluated "Top-1 Element" and "Top-1 Chunk" Transformers, which restrict attention to the most significant token or chunk. The performance of these models aligned with that of the Transformer++, supporting the authors' claims about what is required for generalization.

**Questions:**

The experimental design is sound, and I believe the observations are valuable to the machine learning community. However, the lack of clarity and technical detail are holding the paper back.

- Can you explicitly define base capabilities and specific capabilities?
- Can you provide the exhaustive list of hyperparameters and the pseudo-code for the kernel?

**Ethical Concerns:**

["NO or VERY MINOR ethics concerns only"]

**Final Justification:**

I am pleased with the clarifications. Please disregard my comment on the reproducibility, as the information was available at the time of submission (the appendix was included in the supplementary material). Given the authors’ commitment to including the clarifications and pseudo-code in the paper, I have increased my score from 4 to 5.

**Limitations:**

Yes.

**Paper Formatting Concerns:**

I have no concerns.

**Quality:**

3

**Strengths And Weaknesses:**

This work is timely and relevant considering the recent surge in popularity of alternative architectures and mechanisms to the original Transformer. The authors acknowledge the similarities between their “Top-1 Chunk” attention and Native Sparse Attention, prompting them to shift their focus on extensive ablations and alternative formulations to the Top-1 Chunk attention.

The pre-training methodology is good. The authors pre-trained a broad range of moderately sized models (~110M) on a large dataset (100B tokens from SlimPajama) with a decent sequence length (2k), providing a robust foundation for their observations. The inclusion of larger-scale experiments (1.3B models, 100k context length) and few-shot learning results is appreciated.

Despite its strengths, the paper is limited by two significant weaknesses:
- The central concepts of "base capabilities" and "specific capabilities" are never explicitly defined and are confusing. It seems that “base capability" refers to a model's generalization performance as measured by the ratio of its OOD loss to its ID loss. Note that the OOD set is constructed with (English?) data from domains not included in the train set, such as arXiv and GitHub. The performance is measured on the same task as used for the pre-training. In opposition, "specific capabilities" seems to refer to the performance on downstream tasks. The paper would be substantially improved by explicitly defining these concepts early, and by potentially using more direct phrasing, such as "generalization on the pre-training task".
- The current manuscript lacks the technical details required for reproducibility, including the depth and width of the different models, the optimizer, the scheduler, and all other hyperparameters. The pseudo-code for the custom kernel is also missing.

Minor suggestions:
- Line 33 and 55, I suggest removing “etc” as it is redundant with “such as”.
- Line 64, I believe there is a missing “to” after “needs”.
- In the figures, I suggest adding the identity line to all figures as the x and y-axes are not the same and change between plots.

---

> ### Author Rebuttal · Authors · 2025-07-31
>
> Thanks for your insightful review and valuable feedback!
>
> We answer your questions below.
>
> ---
>
> **Q1:** The central concepts of "base capabilities" and "specific capabilities" are never explicitly defined and are confusing. It seems that “base capability" refers to a model's generalization performance as measured by the ratio of its OOD loss to its ID loss. Note that the OOD set is constructed with (English?) data from domains not included in the train set, such as arXiv and GitHub. The performance is measured on the same task as used for the pre-training. In opposition, "specific capabilities" seems to refer to the performance on downstream tasks. The paper would be substantially improved by explicitly defining these concepts early, and by potentially using more direct phrasing, such as "generalization on the pre-training task" ... Can you explicitly define base capabilities and specific capabilities?
>
> **A1:** We sincerely apologize for the confusion. You are absolutely correct—**we indeed focus on the out-of-distribution (OOD) generalization capability of language modeling as the base capability in our paper**, and we acknowledge the lack of explicit clarification in the earlier sections. We will revise this according to your suggestion.
>
> **Regarding base capabilities,** prior work on stateful sequence modeling primarily evaluated them through in-distribution (ID) generalization capability of language modeling or few-shot learning capability. However, we observed that these metrics fail to adequately distinguish between different sequence modeling architectures. Thus, we propose using OOD generalization of language modeling to measure base capabilities, which has yielded meaningful insights.
>
> **As for specialized capabilities,** we define them as skills such as retrieval, copy, associative recall, and dynamic programming. These are specific capabilities deliberately designed in prior stateful sequence modeling work for architecture analysis, distinct from conventional focuses like language modeling or few-shot learning. Hence, we categorize them as specialized capabilities.
>
> ---
>
> **Q2:** The current manuscript lacks the technical details required for reproducibility, including the depth and width of the different models, the optimizer, the scheduler, and all other hyperparameters. The pseudo-code for the custom kernel is also missing ... Can you provide the exhaustive list of hyperparameters and the pseudo-code for the kernel?
>
> **A2:** We sincerely apologize for any confusion caused. **The technical details (including the depth and width of the different models, the optimizer, the scheduler, and all other hyperparameters) were included in the appendix.** However, the appendix was not embedded in the main paper but instead uploaded as a separate PDF file under **Supplementary Material (the official download link appears at the top of this page)**. Due to the tight deadline for the main paper submission (the Supplementary Material deadline was one week later), we made this arrangement. We will incorporate the appendix into the main paper if given the opportunity in future revisions.
>
> Regarding the kernel implementation, we acknowledge that we only provided the core design diagram without including pseudo-code, and we will follow your suggestion to add it (we attempted several pseudo-code display solutions, such as introducing relevant LaTeX packages or inserting formulas in Markdown code blocks, but **OpenReview doesn't fully support these features**. Therefore, we can only update this in the PDF version later).
>
> Meanwhile, **we are committed to releasing the kernel source code**, which has already been uploaded as **Supplementary Material (the official download link appears at the top of this page)**. We believe the executable source code will help compensate for any insufficient description of kernel details in the paper.
>
> ---
>
> **Q3:** Line 33 and 55, I suggest removing “etc” as it is redundant with “such as” ... Line 64, I believe there is a missing “to” after “needs” ... In the figures, I suggest adding the identity line to all figures as the x and y-axes are not the same and change between plots.
>
> **A3:** Thank you very much for your valuable suggestions regarding the improvement of expressions in our paper. We will carefully revise the manuscript according to your comments.
>
> ---

---

> > ### Comment · Reviewer_zo9Q · 2025-08-05
> >
> > Thank you for the clarifications, they address my first concern about the nature of the base and specific capabilities. My apologies for missing the supplementary material, they address my second concern about the reproducibility. With the authors’ commitment to including the clarifications and pseudo-code in the paper, I’m happy with the rebuttal and will increase my score accordingly.

---

### Official Review · Reviewer_jJs2 · 2025-07-09

**Clarity:** 2
**Significance:** 2
**Originality:** 1
**Rating:** 4
**Confidence:** 4

**Summary:**

This paper investigates non-transformer architectures developed in the past several years to determine which architectural design decisions they make drive their performance. It argues that limited domain ablation experiments - instead of the general purpose / mixed domain pretraining that's typically used - is important for ablation studies. They explore various design decisions made in Mamba and RWKV, as well as introduce and validate two simplifications of ideas common in non-transformer architectures and show that they work.

**Questions:**

1. Why does Section 4 mostly focus on Mamba and Section 5 mostly on RWKV?
2. If one specifically wants to use a non-transformer architecture, what would you recommend they use? I think this is a question a lot of people have.
3. How robust are these results to changing the training dataset?

**Ethical Concerns:**

["NO or VERY MINOR ethics concerns only"]

**Final Justification:**

I find the response to my questions about justifying their dataset decisions and what architecture they'd recommend wholly unsatisfying.

**Limitations:**

Yes

**Quality:**

3

**Strengths And Weaknesses:**

This is a good paper full of careful experiments. My main hesitency is around the fact that some of the experimental design choices seem poorly motivated in ways that could be the result of cherrypicking. To be clear I'm not saying I think the authors did, but that ad hoc design choices are non-desirable and one of the main reasons is that they can enable cherrypicking. The two examples that stand out the most are:
1. Section 4 focuses on comparing to Mamba and Section 5 focuses on comparing to RWKV. It's not clear why this is. The sections are titled "Top-1 Element Selection Architecture" and "Top-1 Chunk Selection Architecture" which doesn't obviously connect to one vs the other.
2. Very little space is spent justifying the decision to use a single data source to do the ablation studies. I would like to hear more about both the decision-making process and results on other datasets. For example, it could be that the specific dataset used is a major driver of the results. Alternatively, I could see people wondering if the results matter if they don't replicate on the types of training data people typically use.
These factors make it somewhat challenging to assess the significance of the results, as they make me wonder about the generality.

The idea of doing an ablation study of architecture design is not at all novel and entirely standard. Nobody should read the previous sentence as being a knock against this paper though.

Many of the plots are unreadable in their current form due to small fonts and sizes. A particularly bad case is Figure 3, where at 250x scale I still can't read the key.

---

> ### Author Rebuttal · Authors · 2025-07-31
>
> Thanks for your insightful review and valuable feedback!
>
> We answer your questions below.
>
> ---
>
> **Q1:** Section 4 focuses on comparing to Mamba and Section 5 focuses on comparing to RWKV. It's not clear why this is. The sections are titled "Top-1 Element Selection Architecture" and "Top-1 Chunk Selection Architecture" which doesn't obviously connect to one vs the other ... Why does Section 4 mostly focus on Mamba and Section 5 mostly on RWKV?
>
> **A1:** We apologize for any confusion caused—this may have been a misunderstanding, and we would like to clarify the following points:
>
> In fact, most of the experiments in Section 4 (Top-1 Element Selection Architecture) and Section 5 (Top-1 Chunk Selection Architecture) are consistent, as both involve comparisons with the Mamba series of models. **Figures 6, Table 1, and Table 2 in the paper are shared between Sections 4 and 5**, presenting the results of comparing the three architectures proposed in these two sections with Mamba-1/2 and others. Due to an oversight in layout, these results were placed closer to Section 4, which may have misleadingly suggested they belonged exclusively to that section. We will improve this in subsequent versions.
>
> Regarding the experimental design related to RWKV-7, the situation is somewhat complex, and we would like to explain further:
>
> 1. RWKV-7 was indeed not included as an independent baseline in Figures 6, Table 1, and Table 2 (i.e., the three architectures from Sections 4 and 5 were not compared with RWKV-7 in these figures/tables, though they were all compared with Mamba-1/2). This is because RWKV-7 incorporates many additional design elements beyond its sequence modeling architecture, which significantly improve its convergence speed. Since these enhancements are not attributable to improvements in the sequence modeling architecture itself, a direct comparison would be unfair and could obscure our assessment of the sequence modeling architectures' merits. Therefore, RWKV-7 was excluded from Figures 6, Table 1, and Table 2.
>
> 2. To address this limitation, **we designed additional experiments in Figure 7 and Table 3**. The core idea here was to replace RWKV-7's original sequence modeling component with different sequence modeling architectures to enable a fair comparison with the original RWKV-7. Since the experiments in Figures 6, Table 1, and Table 2 had already demonstrated that the architecture in Section 4 was weaker in convergence compared to the two architectures in Section 5, we only transplanted the latter two architectures into RWKV-7 for these experiments—primarily to avoid unnecessary computational costs.
>
> To summarize, we did not deliberately design Sections 4 and 5 to compare against different baselines. A more accurate description would be:
>
> **After comparing the architectures in Section 4 and Section 5 with Mamba-1/2, we determined that the two architectures in Section 5 were superior. We then transplanted the two architectures into RWKV-7 and compared them with the original RWKV-7.**
>
> ---
>
> **Q2:** Very little space is spent justifying the decision to use a single data source to do the ablation studies. I would like to hear more about both the decision-making process and results on other datasets. For example, it could be that the specific dataset used is a major driver of the results. Alternatively, I could see people wondering if the results matter if they don't replicate on the types of training data people typically use. These factors make it somewhat challenging to assess the significance of the results, as they make me wonder about the generality ... How robust are these results to changing the training dataset?
>
> **A2:** We appreciate your concerns and would like to provide the following clarification:
>
> All our architecture ablation analyses were conducted during the pre-training phase. Therefore, the "training dataset" you mentioned primarily refers to the pre-training dataset. Pre-training datasets have a distinctive characteristic compared to task-specific datasets: **most publicly available pre-training datasets likely share similar original data sources.** Tracing back to their origins, they are essentially crawled from the public internet, differing mainly in collection scale and data processing methods.
>
> For instance, the SlimPajama dataset we employed consists mainly of data from CommonCrawl, C4, GitHub, Books, ArXiv, Wikipedia, and StackExchange. **The CommonCrawl and C4 subsets we used for pre-training are also core components of many public pre-training datasets** (either directly referenced or with highly overlapping sources), which was the primary reason we selected them for pre-training.
>
> Given such intrinsic connections among different pre-training datasets, we chose not to switch between various pre-training datasets. Instead, we conducted relevant experiments using SlimPajama, which has already undergone cross-subset deduplication. **We believe that** selecting other pre-training datasets while performing similar cross-subset deduplication would yield largely comparable results in architecture ablation analyses.
>
> ---
>
> **Q3:** Many of the plots are unreadable in their current form due to small fonts and sizes. A particularly bad case is Figure 3, where at 250x scale I still can't read the key.
>
> **A3:** We sincerely apologize for any inconvenience caused during the reading process. We fully appreciate your suggestions and will make improvements to the presentation of figures, tables, and text in the revised version.
>
> ---
>
> **Q4:** If one specifically wants to use a non-transformer architecture, what would you recommend they use? I think this is a question a lot of people have.
>
> **A4:** While we hold our work in high regard, it has yet to undergo large-scale validation and thus may not currently represent the most robust option.
>
> Regarding non-Transformer architectures, our primary recommendation remains **Mamba-2**, as it demonstrates superior developmental momentum and maturity. Additionally, we would suggest considering **RWKV-7**, which incorporates numerous unique design elements that significantly enhance convergence (though many are engineering tricks unrelated to sequence modeling, their practical effectiveness is remarkable). Its block design—featuring separated MHA and FFN components similar to Transformers—makes it notably more amenable to customization or reintegration of recent advancements from Transformer architectures.
>
> ---

---

> > ### Comment · Reviewer_jJs2 · 2025-08-08
> > **This response has not increased my confidence in the paper**
> >
> > **Q1:** This makes sense. The sentence "After comparing the architectures in Section 4 and Section 5 with Mamba-1/2, we determined that the two architectures in Section 5 were superior. We then transplanted the two architectures into RWKV-7 and compared them with the original RWKV-7" makes a lot more conceptual sense that what I thought was going on, and I hope you consider how to make this process clearer with the structure of the paper. It may also be worth explicitly stating that sentence at the beginning of the paper.
> >
> > **Q2:** This is non-responsive to my question. A major thing you emphasize is how using mixed domain datasets produces different results than using limited domain ones. Since virtually everyone pretrains on mixed domain datasets, that seems to undermine the value of this study compared to one conducted using a more realistic training mix. At no point do you present evidence that what is optimal under your set-up will also be optimal under realistic training data distributions.
> >
> > **Q3:** Great!
> >
> > **Q4:** If the lack of large scale validation makes you hesitate to make recommendations, isn't that a strike against this paper? If this paper accomplishes anything, I would hope it would provide information about which architectural design choices are better and which are worse.

---

> > > ### Author Response · Authors · 2025-08-08
> > >
> > > **A4:** We sincerely apologize if our wording caused any confusion, and we appreciate the opportunity to clarify our position:
> > >
> > > Our decision not to recommend the architecture was not due to lack of confidence in its design. In fact, we would adopt this architecture ourselves if given the opportunity for large-scale pre-training. However, as is well known, large-scale pretraining involves significant risks. While we are willing to bear such risks ourselves or when others make informed decisions to do so, we prefer not to have others assume these risks based solely on our recommendation. **This reflects our ethical considerations rather than technical reasons.**
> > >
> > > Regarding the comparative analysis of architecture merits, **we compared stateful sequence modeling architectures as a whole with our proposed architecture** to arrive at the conclusions presented in the paper. We did not devote substantial discussion to comparisons among different stateful sequence modeling architectures themselves for two main reasons: First, as these serve as baseline methods rather than the focus of our work, we were concerned that excessive emphasis might obscure the paper's core contributions. Second, since we did not conduct extensive detailed comparisons between various stateful sequence modeling architectures (which was beyond this work's scope), we refrained from making definitive claims about their relative merits. That said, **within the scope of this study, Figure 1(b) does provide relevant comparative information** that we believe will serve both you and other readers well.
> > >
> > > ---

---

> > > > ### Comment · Reviewer_jJs2 · 2025-08-09
> > > > **This rebuttal was worse than not rebutting**
> > > >
> > > > I find this reply utterly unconvincing. Whether it's deliberate or not, the fact that you're failing to engage with a basic question about your data choices (what justifies using a setting that you agree differs from real settings?) worries me greatly. This is a very important point, and saying "well there's no different in perf in the contexts these models are actually used in so we look at a different context" does not instill confidence that your results are meaningful. I was trying to get you to explain why you expect your results to transfer to the standard pretraining set-up.
> > > >
> > > > On comparing non-transformer architectures, of course you can be expected to make recommendations about what non-transformer architecture to use! Studying them is the entire point of this paper! You don't have an ethical duty to not comment on the implications of your work!
> > > >
> > > > This exchange has actively decreased my confidence in the authors. However I don't think it decreases my satisfaction enough to drop my rating. Therefore I reluctantly leave my score unchanged.

---

> ### Author Response · Authors · 2025-08-08
>
> Thanks for your insightful response!
>
> We will continue to answer your questions below.
>
> ---
>
> **A1:** Thank you for your valuable feedback! We will carefully consider your suggestions and make targeted improvements in the subsequent version of our manuscript.
>
> ---
>
> **A2:** We sincerely apologize for our previous misunderstanding of your question. Please allow us to provide a clarified explanation:
>
> This issue is somewhat similar to the first question. In fact, we presented the results of mixed domain pre-training in the first experiment of our paper (**Figure 1(a)**). We observed that under this setting, various sequence modeling architectures achieved similar language modeling test performance when reaching comparable pre-training levels. This led us to conclude that under the mixed domain pre-training setting with in-distribution test, it might be difficult to distinguish between the merits of different architectures. Consequently, we proceeded to explore the use of out-of-distribution test settings, which we subsequently adopted for all following experiments.
>
> When implementing the out-of-distribution test settings, **we considered two approaches:** 1) maintaining the mixed domain pre-training setting while seeking new, unused domain data for test; or 2) adopting a limited domain pre-training setting where certain domains are reserved exclusively for test. As you rightly pointed out, the first approach better aligns with real-world scenarios. However, its main challenges lie in collecting additional new domain data, and these new domains might not be widely familiar, potentially appearing artificially selected. The second approach, while differing from mainstream pre-training methods, is more practical to implement and ensures that out-of-distribution domains are well-known. Moreover, since this setting was designed specifically for architecture analysis, it doesn't conflict with or replace large-scale mixed domain pre-training for practical applications. **Based on these considerations, we chose the limited domain pre-training setting to achieve our out-of-distribution test objectives.** However, as you rightly pointed out, this approach comes at the cost of losing the opportunity to directly validate the architecture's OOD generalization capability under mixed domain pre-training settings. That said, we believe out-of-distribution generalization is a universal challenge—mixed domain pre-training can hardly cover all possible domains, making it somewhat similar to limited domain scenarios in this regard. We hope to further explore and validate this aspect in future work.
>
> We understand your additional concern regarding our lack of demonstration of the proposed architecture's performance under mixed domain pre-training. This primarily relates to our early decision to adopt the limited domain pre-training setting after the first experiment. However, during the preliminary stages of this work, we did conduct experiments with the Top-1 Element Selection architecture under mixed domain pre-training. We present these results below:
>
> **(1) When Model parameters≈110M: Language modeling test performance under mixed domain pre-training**
>
> As we cannot insert a figure here, we present some data points where the pre-training levels can be aligned in a tabular format for your reference (**analogous to observing the differences in vertical coordinate values while fixing the horizontal coordinate values in Figure 1(a)**) :
>
> | Model | Pre-Training Loss | Loss on Mixed Domain Test Set (ID) |
> | :-----| :-----: | :-----: |
> | Transformer++ | 2.83619 | 2.30881 |
> | Top-1 Element | 2.83130 | 2.30267 |
> | | | |
> | Transformer++ | 2.79629 | 2.26666 |
> | Top-1 Element | 2.79794 | 2.26627 |
> | | | |
> | Transformer++ | 2.77541 | 2.24591 |
> | Top-1 Element | 2.77463 | 2.24285 |
> | | | |
> | Transformer++ | 2.75492 | 2.22381 |
> | Top-1 Element | 2.75587 | 2.22106 |
>
> **(2) When Model parameters≈1.3B: Language modeling test performance under mixed domain pre-training**
>
> Similarly, we present these results in tabular form:
>
> | Model | Pre-Training Loss | Loss on Mixed Domain Test Set (ID) |
> | :-----| :-----: | :-----: |
> | Transformer++ | 2.39902 | 1.88991 |
> | Top-1 Element | 2.39882 | 1.89060 |
> | | | |
> | Transformer++ | 2.35962 | 1.85101 |
> | Top-1 Element | 2.36057 | 1.85379 |
> | | | |
> | Transformer++ | 2.33047 | 1.82144 |
> | Top-1 Element | 2.33449 | 1.82770 |
> | | | |
> | Transformer++ | 2.30021 | 1.79239 |
> | Top-1 Element | 2.30367 | 1.79699 |
>
> Since we observed that the behavior of the **Top-1 Element Selection** architecture in the mixed domain pre-training setting **remained consistent with our earlier observations (Figure 1(a)) without showing degradation**, we conducted only unified limited domain pre-training experiments for the Top-1 Chunk Selection architecture, which shares similar mechanisms. We will incorporate these results into subsequent versions of the paper to further strengthen its persuasiveness.
>
> ---

---

### Author Response · Authors · 2025-08-09

We sincerely appreciate the time and effort that all reviewers have dedicated to evaluating our work! Your valuable suggestions have been instrumental in enhancing the clarity and completeness of our paper. We are truly grateful for your insightful feedback and support!

---

### Decision · Program_Chairs · 2025-09-17

**Decision:**

Accept (poster)

**Comment:**

This paper studies how neural architecture choices affect the base capabilities of LLMs. The core claim is that mixed-domain pretraining often hides architectural differences early, and that limited domain pre-training setting with out-of-distribution testing is a more discriminative way for probing. With this setup, the authors find that stateful architectures (lie Mamba/RWKV) show notable degradation relative to Transformers. They also propose a design principle:  "a sequence modeling architecture need possess full-sequence arbitrary selection capability".
They validate this by introducing two intentionally simple Top-1 Transformers, which basically restricts attention to the most significant token or chunk. The Top-1 Transformers can match Transformer++ on OOD generalization at small scale (≈110M params, 2k context) and remain competitive at larger scale (≈1.3B, 100k context).

Strengths:
1. The topic is important and the conclusion is insightful for LLM pretrain community
2. The FSAS principle is simple, testable, and of high practical relevance for ongoing non-Transformer efforts.
3. Experiment design is good, it controlled training budgets, early-stage OOD evaluations, and scaling checks.

Weaknesses:
1. One major concern is presentation. I feel difficult to understand the paper at my first try. Especially when the authors tried to introduce the top-1 models. Many sections begin with the lack of motivations.
2. The main conclusions rely on a limited-domain pretraining + OOD evaluation protocol. Although authors claimed that mixed-domain pretraining creating the illusion of base capabilities. It still raised the question that how generalizable the conclusion of this setting can be applied to real-world LLMs.

Rationale for Accept:
Even if majority community members continue to favor mixed-domain pretraining, the presented framework and results are still valuable to architecture ablation studies. Also, two reviewers increased their scores following clarifications and added results.